



**Two years online measurement of fine particulate nitrate in western**
**Yangtze River Delta: Influences of thermodynamics and N₂O₅ hydrolysis**
Peng Sun[1], Wei Nie[1,2,*], Xuguang Chi[1,2], Yuning Xie[1], Xin Huang[1,2], Zheng Xu[1,2], Ximeng Qi[1,2], ,
Zhengning Xu[1], Lei Wang[1], Tianyi Wang[1], Qi Zhang[3], and Aijun Ding[1,2,*]
[1]Joint International Research Laboratory of Atmospheric and Earth System Science s, and School of
Atmospheric Sciences, Nanjing University, Nanjing, 210023, China
[2]Jiangsu Provincial Collaborative Innovation Center of Climate Change, Nanjing, 210023, China
[3]Department of Environmental Toxicology, University of California, Davis, CA 95616, USA
Correspondence: Wei Nie (niewei@nju.edu.cn) and Aijun Ding (dingaj@nju.edu.cn)
**Abstract.**
Particulate nitrate contributes a large fraction of secondary aerosols. Despite
understanding of its important role in regional air quality and global climate, long-term
continuous measurements are rather limited in China. In this study, we conducted online
measurement of PM$_{2.5}$ nitrate for two years from March 2014 to February 2016 using
the Monitor for Aerosols and Gases in ambient Air (MARGA) in the western Yangtze
River Delta (YRD), eastern China, and investigate the main factors that influenced its
temporal variations and formation pathways. Compared to other sites in China, an
overall high concentration of particulate nitrate was observed with a mean value of 15.8
µg m$^{-3}$ (0.5 to 92.6 µg m$^{-3}$). Nitrate on average accounted for 32% of the total mass of
water-soluble ions and the proportion increased with PM loading, indicating that nitrate
is a major driver of haze pollution episodes in this region. Sufficient ammonia drove
most nitrate into the particle phase in the form of ammonium nitrate. A typical seasonal
cycle of nitrate was observed with the concentrations in winter on average two times
higher than those in summer mainly due to different meteorological conditions. In
summer, the diurnal variation of particulate nitrate was determined by the
thermodynamic equilibrium, resulting in a much lower concentration during daytime
despite of a considerable photochemical production. Air masses from polluted YRD



and biomass burning region contributed to the high nitrate concentration during summer.
In winter, particulate nitrate didn't reveal an evident diurnal variation. Regional
transport from northern China played an important role in enhancing nitrate
concentration. Eighteen nitrate episodes were selected to understand the processes that
drive the formation of high concentration of nitrate. Rapid nitrate formation was
observed during the pre-episode (the day before nitrate episode day) nights, and
dominated the increase of total water-soluble ions. Calculated nitrate from $N_2O_5$
hydrolysis was highly correlated to and accounted for 80 percent of the observed nitrate,
suggesting that $N_2O_5$ hydrolysis was a major contributor to the nitrate episodes. Our
results suggested that rapid formation of nitrate could be a main cause for extreme
aerosol pollution events in YRD during winter, and illustrated the urgent needs to
control the $NO_x$ emission.
**1.   Introduction**
Particulate nitrate ($NO_3^-$), as a major aerosol component in the atmosphere, reduces
atmospheric visibility (Charlson and Heintzenberg, 1995), influences human health,
alters radiative forcing and hence influences regional even global climate (IPCC, 2013).
Compared to the sulfate, nitrate has a larger scattering albedo under low RH conditions
that cause a stronger influence on visibility (Lei and Wuebbles, 2013). High
concentration of particulate nitrate had been demonstrated to be one of the major
reasons for the frequent occurrence of haze episodes in China (Wang and Zhang, 2009;
Wen et al., 2015; Wang et al., 2017). In recent decades, Chinese government started to
control emissions of air pollutants with special effort on the $SO_2$ reduction. This resulted
in a remarkable decrease of ambient $SO_2$ and sulfate concentrations after 2006 (van der
A et al., 2017; Wang et al., 2017). However, particulate nitrate, as well as its proportion
in PM, showed increasing trends due to the strong emission of nitrogen oxides ($NO_x$)
(Lei and Wuebbles, 2013; Yang et al., 2017).
Particulate nitrate can be formed from multiple pathways. Gas phase reaction of
$NO_2$ and OH radical is one major pathway to form nitric acid ($HNO_3$) (Calvert and



Stockwell, 1983), which subsequently reacts with ammonia ($NH_3$) to produce
ammonium nitrate ($NH_4NO_3$). As typical photochemical processes, these reactions
dominate daytime nitrate formation, and have been widely investigated in both field
and modelling studies (Sharma et al., 2007; Petetin et al., 2016). Heterogeneous uptake
of the photochemical formed nitric acid by alkali compounds, e.g. dust and sea salt
particles, is also a considerable pathway to form nitrate in some regions (Bian et al.,
2014). During nighttime, the hydrolysis of dinitrogen pentoxide ($N_2O_5$) is believed to
be the dominate pathway to form particulate nitrate. $N_2O_5$ is an important reactive
nitrogen species in the polluted troposphere (Brown and Dube., 2007; Osthoff et al.,
2006; Li et al., 2017; Brown et al., 2003; Brown and Stutz, 2012) and accumulates via
the reversible reaction between $NO_2$ and $NO_3$ radical produced from the reaction of
$NO_2$ with $O_3$. Due to the rapid photolysis of $NO_3$ radical, $N_2O_5$ concentration during
daytime is rather low that its contribution to nitrate can be ignored. While during
nighttime, $N_2O_5$ concentration can be up to ppb level, and form nitric acid by reaction
with water vapor, or particulate nitrate directly by heterogeneous hydrolysis on the wet
surface (Wang et al., 2017; Wen et al., 2018; Thornton et al., 2003). In China, the
pollution episodes with high nitrate concentrations mostly occurred in winter, during
which the photochemical production of nitrate should be overall weak. $N_2O_5$ hydrolysis
thus has the potential to be the crucial contributor, however there is still a lack of
observational evidences.

Collecting particulate matter on a filter with subsequent ion chromatography

analysis in laboratories is the conventional method to measure the concentration of
particulate nitrate. The un-denude filter pack system, which is most-widely used, can
suffer from both positive artifacts by absorbing gas-phase nitric acid, and negative
artifacts by the evaporation of ammonium nitrate (Nie et al., 2010; Pathak and Wu,
2009; Wang et al., 2010). A denuder system can minimize these sampling artifacts by
adding denuders to remove the interfering gases and back-up filters to collect the
evaporated vapers (John et al., 1988). However, the operation of such a denuder system





is super labor intensive and thus not widely used. In addition, the poor time-resolution
of filter-based measurement can limit our understanding on the formation and chemical
evolution of the particulate nitrate. To overcome these shortcomings, several
continuous and semi-continuous techniques have been developed based on an online
denude-IC system (e.g. the ambient ion monitor (AIM), the gas and aerosol collector
ion chromatography (GAC-IC), the particle-into-liquid sample ion chromatography
(PILS-IC) and MARGA), as well as mass spectrometry (e.g. AMS). Pathak et al. (2011)
applied an AIM instrument in Beijing and Shanghai for one month and found that the
heterogeneous hydrolysis of $N_2O_5$ contributed 50%~100% of the nighttime nitrate
formation. Xue et al. (2013) deployed a PILS-IC system in Hong Kong for less than a
month and showed a more active nitrate formation during PM episode than normal days.
Wen et al. (2015) used a MARGA instrument in Yucheng, North China during summer
and emphasized the important roles of $O_3$ and $NH_3$ on nitrate formation. Yang et al.
(2017) carried out field observation with ACSM in Beijing for half a month and pointed
out the importance of aerosol nitrate in haze formation. However, despite of an
increasing number of studies using online techniques, continuous measurements with
more than one-year period are still very limited.

The Yangtze River Delta (YRD) located in the eastern China, with megacities

including Shanghai, Nanjing and Suzhou etc., has suffered from heavy particulate
matter pollution and photochemical pollution (Ding et al., 2013bc; Wang et al., 2016bc;
Wang et al., 2016a). Previous studies indicated an important role of nitrate in the
pollution episodes (Hua et al., 2015; Du et al., 2011; Yang et al., 2017). For example,
Zhang et al. (2015) carried out an observation with ACSM in urban Nanjing during
summer and autumn, and found nitrate and organic aerosols dominated the $PM_1$
composition. Shi et all. (2014) used MARGA instrument in Shanghai for months and
found and increasing contribution of nitrate to $PM_1$ mass during pollution periods.
Wang et al. (2016b) reported temporal variation and transport of $PM_{2.5}$ water soluble
ions, including nitrate, in an urban site in Shanghai based on three-year continuous





measurement using MARGA. However, detailed investigation on the possible
mechanisms governing nitrate behaviors during haze pollution is still rare.

In this study, we present a 2-year continuous measurement of particulate nitrate

using MARGA at a rural site in Nanjing, a megacity in the western YRD region, with
the target to get a comprehensive understanding of particulate nitrate behaviors and
investigate the processes affecting nitrate in haze episodes. We first conducted general
statistical analysis of particulate nitrate and characterized seasonal variation and diurnal
pattern. A thermodynamic model was then applied to investigate the gas-particle
partition of nitrate. The influence of air masses was also investigated by conducting
backward Lagrangian dispersion modelling. Finally, we selected eighteen nitrate
episodes and investigated the main processes influencing their evolution.
**2.   Methodology**
2.1.   Sample site and instrumentation
The SORPES station (118°57′E, 32°07′N) was located on the top of a small hill (40 m
above sea level) in the Xianlin campus of Nanjing University located in the outskirts of
Nanjing, China. The station is an ideal receptor of air masses from the YRD with little
influence of local emissions and urban pollution from Nanjing. Detailed description can
be found in previous studies (Ding et al., 2013c; Ding et al., 2016a).

The measurement was conducted from March 2014 to Feb 2016. Hourly

concentrations of water soluble gases of HCl, $HNO_3$, HONO, $SO_2$ and $NH_3$, and water-
soluble ions in $PM_{2.5}$, including $Cl^-$, $NO_3^-$, $SO_4^{2-}$, $NH_4^+$, $Na^+$, $K^+$, $Ca^{2+}$, and $Mg^{2+}$, were
measured with a Monitor for Aerosols and Gases in ambient Air (MARGA, designed
and manufactured by Applikon Analytical B.V., the Netherlands) employed in
connection with a Thermo $PM_{2.5}$ cyclone inlet. The sampling system was comprised of
two parts: A Wet Rotating Denuder for gases and a Steam Jet Aerosol Collector for
aerosols, working at an air flow of 1 $m^3$ $h^{-1}$ (ten Brink et al., 2007; Rumsey et al., 2014).
After each hour's collection, the samples were analyzed using ion chromatography. The
instrument was calibrated on an hourly basis using internal standard liquid (bromide





lithium), ensuring a stable and reliable ion chromatograph. Concentrations of all aerosol
ions and gases have a precision of 0.001μg m$^{-3}$ (Xie et al., 2015). The PM$_{2.5}$ ion dataset
from the MARGA provided more than 15000 hourly samples over the 24 months of
measurements considering points where NO$_3^-$, NH$_4^+$ and SO$_4^{2-}$ were all available. Trace
gases (i.e., O$_3$, SO$_2$, NO$_x$, NO) and PM$_{2.5}$ mass concentrations were also measured at
SORPES (Ding et al., 2013c; Nie et al., 2015; Ding et al., 2016a), together with
meteorological data including wind speed/direction, temperature, and relative humidity.
2.2.   Thermodynamic constants and ISORROPIA II
Formation of ammonium nitrate involves an equilibrium reaction between the gas phase
NH$_3$ and HNO$_3$, and particle phase NH$_4$NO$_3$. The gas-to-particle partitioning is
temperature dependent, and the equilibrium constant can be calculated as follows (units
mol$^2$kg$^{-2}$atm$^{-2}$) (Sun et al., 2011; Seinfeld and Pandis, 2006):
$$K = k_{298}\, exp(a(298/T-1) + b[1 + ln(298/T) - 298/T]) \qquad (1)$$
$$K_{298} = 3.5 \times 10^{16}(atm^{-2}),\ a = 75.11,\ b = -13.5. \qquad (2)$$
The dissociation constant, $K_p$ (T), is the value to examine the product of the partial
pressures of NH$_3$ and HNO$_3$ in ideal equilibrium state at a specific temperature and can
be calculated as follows (units: ppb$^2$) (Sharma et al., 2007; Seinfeld and Pandis, 2006):
$$ln\ K_P = 84.6 - 24220/T - 6.4 ln(T/298)\ \text{(T is temperature in Kelvin)} \qquad (3)$$
Note that the constant is quite sensitive to temperature changes. Lower values of K$_p$
correspond to lower equilibrium values of the NH$_3$ and HNO$_3$ gas phase concentrations,
shifting the equilibrium of the system toward the aerosol phase.

ISORROPIA II (available at http://isorropia.eas.gatech.edu/) is a thermodynamic

model used commonly in inorganic aerosol research (Fountoukis and Nenes, 2007). To
analyze gas-into-particulate pathway for nitrate formation, HNO$_3$ was modeled with
ISORROPIA II run in forward model iteratively (Pusede et al., 2016; Fountoukis and
Nenes, 2007). ISORROPIA II was initialized as [NO$_3^-$+HNO$_3$]$_{total}$ = [NO$_3^-$]$_{aerosol}$.
Calculated HNO$_3$(g) was added back to [NO$_3^-$+HNO$_3$], while we always use
[NH$_4^+$$_{aerosol}$+NH$_3$(g)] as input total ammonium. ISORROPIA II was solved iteratively





until output $NO_3^-$ changed by < 2% by mass. The phase state was set as metastable. We
assume that gases and aerosol are in equilibrium, that aerosols are homogeneous and
internally mixed, and that unaccounted-for factors do not influence the thermodynamics
of system (Vayenas et al., 2005).
2.3.  Lagrangian Dispersion Modeling
To help understand the influence of air masses, backward Lagrangian particulate
dispersion modeling (LPDM) was carried out based on a method developed and
evaluated by Ding (Ding et al., 2013a). The LPDM was conducted using the Hybrid
Single-Particulate Lagrangian Integrated Trajectory model developed in the Air
Resource Laboratory (ARL) of the National Oceanic and Atmospheric Administration
using the ARL format Global Data Assimilation System data. The model calculates the
position of particulates by mean wind and a turbulence transport component after they
are released at the source point for a backward simulation. For each hour, 3000
particulates were released at 100 m altitude over the site and were traced backward for
a 3-day period. The hourly position of each particulate was calculated using a 3-D
particulate, i.e., horizontal and vertical, method. The residence time at 100 m altitude,
i.e., foot-print "retroplume", which represents the distribution of the surface probability
or residence time of the simulated air mass, was used to understand the contribution
from potential source regions (Ding et al., 2013ac; Shen et al., 2018).
2.4.  Steady-state predictions
Based on their short lifetimes, the concentrations of the $NO_3$ radical and $N_2O_5$ can be
predicted by steady-state calculations due to lack of measurement data (Osthoff et al.,
2006). The formation and loss of $N_2O_5$ associated with a series of chemical reactions
are listed in Table 1. For the heterogeneous processes, we used 0.004 and 0.03 as the
uptake coefficients of the $NO_3$ radical and $N_2O_5$ ($\gamma NO_3$ and $\gamma N_2O_5$), respectively
(Aldener et al., 2006; Wen et al., 2015; Knopf et al., 2011; Brown et al., 2006). Due to
the lack of VOC measurement data, the total reaction rates of $NO_3$ with VOCs were
assumed to be equal to the $NO_3$ loss rate caused by the heterogeneous hydrolysis of



N₂O₅ (Aldener et al., 2006; Wen et al., 2015). The corresponding rate constant can be
found in Master Chemical Mechanism (MCM version 3.1,
http://mcm.leeds.ac.uk/MCM/).

**3. Results and discussion**


3.1. Overall results
A MARGA was deployed to continuously measure the eight water-soluble ions (WSI)
of $PM_{2.5}$ and several gas-phase species from March 2014 to February 2016 with the
time resolution of 1-hour. Nitrate ($NO_3^-$), sulfate ($SO_4^{2-}$) and ammonium ($NH_4^+$) were
the major components with the two-year averaged concentrations of 15.8 (±13.4),15.3
(±10.6) and 10.4 (±7.6) µg m⁻³, respectively. In the present study, we focused on nitrate,
and discussed the temporal variation and its association with physicochemical
processes. The concentration of $PM_{2.5}$ nitrate changed largely from 0.5 to 92.6 µg m⁻³
during the measurement period, and accounted for 3% to 58% of total WSI (=45.7 +
30µg m⁻³). The highest hourly nitrate concentration occurred on December 23, 2015
together with high concentrations of sulfate (65.5 µg m⁻³), and ammonium (56.8 µg m⁻³
³). Heavy haze episode like this occurred at our site frequently during winter. To
understand the influence of wind on nitrate concentration, the wind rose plot is given
in Fig. S1. The prevailing winds at the SORPES station were from northeast and east
during the two-year observation period. Particulate nitrate tended to accumulate or
formed under stagnant condition of low wind speed. The wind from west and east can
lead to a higher nitrate concentration and also other aerosol components (Ding et al.,
2013bc; Shen et al., 2018), which may be associated with air masses from biomass
burning region and the city clusters of YRD.
To get an overall picture of nitrate distribution in developed region of costal China,
we reviewed results from available nitrate measurements in three most polluted regions
of North China Plain (NCP), YRD and Pearl River Delta (PRD), and summarized their
concentration, sampling sites and measurement techniques in Fig. 1. Measurements in
summer and winter were separated due to the large seasonal difference of particulate



225 nitrate. Despite that these measurements were from various measurement techniques,

226 the results still can give us some insights about the differences in spatial and temporal

227 scales. First, particulate nitrate generally showed the highest concentration in NCP and

228 followed by YRD and PRD. This was in consistence with the spatial distribution of $NO_2$

229 – a major gas precursor of nitrate. Second, evident seasonal variations can be observed

230 at all three regions with much higher concentrations in winter. Third, there was an

231 overall increase trend of particulate nitrate in NCP and YRD in the past decade,

232 especially that during summertime. Nevertheless, particulate nitrate in PRD revealed

233 an overall decreasing trend. Compared to these previous studies, the nitrate

234 concentration during summertime at SORPES station was lower than that in NCP, but

235 higher than that in YRD and PRD cities. In terms of wintertime, nitrate concentration

236 at SORPES station was slightly lower than that in NCP, comparable with that in YRD,

237 and higher than that in PRD.

238  Fig. 2 illustrates the occurrence frequency of the loading of particle matter in

239 different concentration range, and the changes of nitrate proportion along with the PM

240 loading. Noting that the PM loading here was indicated by the mass of total WSI. The

241 highest frequency of WSI concentrations occurred in a range of 20-40 μg m$^{-3}$, and

242 gradually decreased with the increasing of concentration. Heavy PM pollution with

243 WSI concentrations higher than 100 μg m$^{-3}$ occurred during more than 5% of the time

244 during this study. The contribution of nitrate to total WSI increased with the PM loading,

245 ranging from ~25% with WSI concentration lower than 20 μg m$^{-3}$ to ~ 40% when WSI

246 was higher than 140 μg m$^{-3}$. These results suggested that nitrate was a major driver of

247 haze episodes with high PM peaks in this region.

248  Fig. 3a shows the scatter plot of particulate nitrate and total WSI. They overall

249 correlated to each other with correlation coefficient R = 0.92, and nitrate accounted for

250 32% of the total WSI. Air temperature greatly affected the contribution of nitrate to total

251 WSI. Its proportion can be up to 58% at around 0 °C and only 3% at the temperature

252 higher than 30 °C, indicating an important role of thermodynamic equilibrium in nitrate





concentration. We further investigated the neutralization extent of sulfate and nitrate by
ammonium (Fig. 3b). Ammonium was overall enough to neutralize both sulfate and
nitrate, suggesting that the particulate nitrate mostly existed as ammonium nitrate at
SORPES station, contrasts with some ammonia poor regions, where excess $NH_4^+$
defined as $([NH_4^+]/[SO_4^{2-}]-1.5) \times [SO_4^{2-}]$ played an important role in the formation of
particulate nitrate (Pathak and Wu, 2009). Evident seasonal difference can be observed
for the molar ratio of ammonium to the sum of sulfate and nitrate. In spring and early
summer, a fraction of the particulate nitrate is present in the forms of $Ca(NO_3)_2$ and
$KNO_3$; while in winter, considerable chloride would consume some ammonium to form
$NH_4Cl$.
3.2. Characteristics of fine particular nitrate in different seasons.
3.2.1. Seasonal pattern and its main causes
Fig. 4 shows the composite seasonal pattern of $NO_x$, $PM_{2.5}$ nitrate, sulfate and the ratio
of nitrate to sulfate during the 2-year period at SORPES station. Similar to the previous
studies (Griffith et al., 2015), a typical seasonal variation was observed for particulate
nitrate (and its ratio to sulfate, i.e. $NO_3^-/SO_4^{2-}$), with a maximum value of 23.7 µg m$^{-3}$
(140%) in January, and a minimum of 8.4 µg m$^{-3}$ (66%) in August and September.
Particulate sulfate revealed a bimodal pattern with high concentrations occurred in
January and June, respectively. The low value of particulate nitrate during summer can
be generally explained by the higher temperature, higher and unstable boundary layer
and relative clean air masses induced by summer monsoon (Ding et al., 2013c) despite
of the increased photochemical formation. In opposite, the high values during winter
were generally due to the lower temperature, lower and stable boundary layer and
relative stronger continental outflow from the North China where anthropogenic
emission was relatively high due to heating in winter (Ding et al., 2013c). Different
chemical processes that affects nitrate concentrations between summer and winter will
be discussed later. $NO_x$, the major precursor, tracked the changes of particulate nitrate
commendably, except for that during February and June. In addition, a secondary peak



of particulate nitrate can be observed during June, which can be explained as the
influence from agricultural burning in eastern China (Ding et al., 2013bc; Xie et al.,
2015; Shen et al., 2018). The concentrations of Potassium, a biomass burning tracer in
this region (Ding et al., 2013b; Xie et al., 2015), clearly showed a consistent peak (Fig.
S2) with both particulate nitrate and sulfate, as well as the discrepancy of $NO_x$ and
nitrate concentrations (Ding et al., 2013b; Xie et al., 2015; Nie et al., 2015) shown in
Fig. 4. While in February, the nitrate concentration didn't show concurrent decrease in
$NO_x$ during the Chinese Spring Festival (Ding et al., 2013c). It might suggest that local
emissions were not the major contributor to the particulate nitrate observed in winter.

Thermodynamics is an important factor influencing the formation and partitioning

of nitrate. Here, we calculated the dissociation constant ($K_p$) to evaluate whether
ambient conditions favored the formation of aerosol $NH_4NO_3$. The hourly $NH_3$
concentration was measured by MARGA and $HNO_3$ was calculated by ISORROPIA II.
As showed in Fig. 5, the product of $[NH_3(g)]$ and $[HNO_3(g)]$ for most ambient samples
during summer was lower than the calculated $K_p$, suggesting ambient condition during
summer at SORPES favored to evaporate ammonium nitrate to gas phase $NH_3$ and
$HNO_3$ (Fig. 5a). In contranst, ambient conditions during winter favored to form
particulate nitrate in most cases (Fig. 5b). This study also shows the similar result with
the study reported in the Kanpur, India (Sharma et al., 2007).
3.2.2.  Diurnal cycles during summer and winter
In Fig. 6, we show the averaged diurnal variations of particulate nitrate, nitrogen
dioxide, nitric acid, equilibrium constant ($K$), air temperature and RH during summer
and winter during the two years. Nitric acid was calculated by ISORROPIA II. In
summer (Fig. 6a), the fine particulate nitrate showed a typical diurnal cycle that the
maximum concentration occurred at 7:00 with the average concentration of 16.5 µg m⁻
³ and minimum value at 14:00 (7.2 µg m⁻³). This summertime diurnal pattern of nitrate
is very similar with the findings in Shandong (Wen et al., 2015) and New York (Sun et
al., 2011). However, it is quite different from the findings in Hong Kong (Griffith et al.,



2015), where nitrate concentration peaks in the daytime in summer. Ambient
temperature and the development of boundary layer are the major drivers to the
observed diurnal variation of particulate nitrate, and high temperature and high
boundary layer during daytime prefer to evaporate and dilute the particulate nitrate
(Zhang et al., 2015a; Ding et al., 2016). Nitric acid, which accounted for 20% of the
total nitrate [$NO_3^-$+$HNO_3$], revealed its high concentration (around 2 ppb) in the
noontime (12:00-15:00). $NO_2$, the precursor of nitrate, showed a peak concentration of
18.2 ppb at 21:00, and remained at a high level during the whole night. Equilibrium
constant, $K$, was calculated to understand the influence of gas-to-particulate partitioning
on the observed diurnal variation of particulate nitrate (Sun et al., 2011). As showed in
Fig. 6a, $K$ was highly correlated to particulate nitrate, suggesting the thermodynamic is
the major factor influencing the diurnal variation of particle nitrate during summer.

In winter, the diurnal variation is small with a considerable peak appeared at

around 10:00 AM. Compared to that in summer, K showed similar diurnal variation,
but not correlated to particulate nitrate, indicating factors other than the control of
temperature. The observed peak at late morning was probably due to    downward
mixing from the residual layer where particulate nitrate was formed aloft during the
night and brought to the surface after sunrise following the breakup of the boundary
layer (Brown and Dube., 2007; Young et al., 2016; Pusede et al., 2016). Direct vertical
observations are needed to further investigate this issue.

To further investigate factors influencing the nitrate behaviors other than

thermodynamics, ISORROPIA II was used to simulate the diurnal variation of nitrate.
Hourly concentrations of all species (both gas and aerosol phase species) at 00:00 were
used as the initial value of each specific day. Hourly data of temperature and relative
humidity were used as the input data to constrain the model. The ISORROPIA II model
was set as forward mode and metastable phase state. The calculated diurnal variations
were showed in Fig. 7 together with the observed results.

The differences between the calculation and the observation could be attributed to:





(1) the development of boundary layer, (2) the dry deposition of nitric acid, and (3)
chemical processes, which has not been considered yet in the model. As shown in Fig.
7a, the overall diurnal pattern of nitrate in summer is well captured by the model except
for 3 periods. The differences after midnight are likely caused by the effect of boundary
layer height and some chemical processes. Faster increase of model nitrate after 18:00
was attributed to lack of dry deposition of nitric acid in the model. During noontime the
observed nitrate concentration was expected to be lower than the calculated value
because of the development of boundary layer and stronger dry deposition of nitric acid
associated with stronger turbulence mixing, which were ignored in the model. However,
in contrast, the observation was considerably higher than the calculated value. It
indicates a strong production of nitrate via photochemical processes in summer. Fig.7c
shows that the difference between calculated and observed nitrate concentration was in
good correlation with the production of $NO_2$ and solar radiation, a proxy for production
rate of nitric acid (Zhang et al., 2005; Young et al., 2016), further suggesting that photo-
oxidation of $NO_2$ is an important source of nitrate during summer, even though the
thermodynamic equilibrium is the dominate factor controlling the diurnal cycle. Wen et
al. (2018) also demonstrated that photochemical production of nitric acid is a major
contributor to daytime nitrate increase during summer in North China Plain. In winter,
the influence of thermodynamics is small since the low temperature. The peaks in the
morning may be caused by the mixing down of a residual layer enriched of nitrate as
mentioned above, while the decline during the afternoon are supposed to be the result
of dilution associated with the boundary layer development.
3.2.3 Influence of air masses transport
Meteorological processes play a key role in air masses long-range transport and local
accumulation (Ding et al., 2013ac; Zhang et al., 2016; Ding et al, 2016b). In order to
investigate the influence of air masses transport on nitrate concentrations, Lagrangian
dispersion modeling was conducted for the sampling days with the highest and lowest
25[th] percent nitrate concentration in summer and winter, respectively (Ding et al.,



2013a). Fig. 8 shows the retroplumes, i.e. footprint at an altitude of 100 m, of the
selected days during summer and winter, respectively. In summer, high concentrations
of nitrate tended to accompany with the air masses from west of Nanjing (mostly Anhui
province) and Yangtze River Delta (Suzhou-Shanghai city clusters and North Zhejiang
province) (Fig. 8a). YRD is a high $NO_x$ emission region (Fig. 1), air masses from which
could bring high concentration of $NO_x$ to enhance the nitrate concentration at SORPES
station. Biomass burning is the possible cause of the high nitrate loading with air mass
from the west of Nanjing (Fig. S3). In winter, regional transport from northern China
played an important role in enhancing nitrate concentrations. As shown in Fig. 8c, a
large part of air masses for the highest 25% sampling days was from North China Plain,
which has the strongest $NO_x$ emission in China (Fig. 1). It should be noted here that the
longer lifetime of particulate nitrate during winter might be the main cause to promote
the contribution of regional transport to the observed nitrate at SORPES. In contrast,
the lowest 25% sampling days during winter tended to be accompanied with the air
mass from Nanjing local and marine areas.
3.3.   Contribution of $N_2O_5$ hydrolysis to nitrate episodes
Similar to findings from previous studies (Zhang et al., 2015c), nitrate was found to
increase significantly during this study and become the largest contributor of $PM_{2.5}$
during the haze episodes (Fig. 2). Generally, these pollution episodes mainly occurred
in winter (Fig. 3a and Fig. 4), during which the photochemical production of nitric acid
should be weak. $N_2O_5$ hydrolysis was thus proposed to be a potential important
formation pathway. Here we investigated the nitrate episodes in detail and discussed
their relationship to the $N_2O_5$ hydrolysis during the nights before.

In Fig. 9, we show a typical case of nitrate episodes from 30 November to 2

December, 2015. Fast nitrate formation was observed, which was likely caused by
hydrolysis of $N_2O_5$. Nitrate increased significantly from 20.3 µg m$^{-3}$ at 18:00 of 30
November to 63 µg m$^{-3}$ at 6:00 of 1 December, 2015. The ratio of nitrate to $PM_{2.5}$ also
exhibited a large increase from 25% at 18:00 to 38% at 06:00. In contrast, other $PM_{2.5}$





components, e.g. sulfate and black carbon, showed only slight increases. High
concentration of $NO_2$, considerable level of $O_3$ and extremely low concentration of NO
provided a favorable condition towards forming $NO_3$ and $N_2O_5$ (Brown et al., 2003).
The meteorological conditions during these 12 hours were stable with low wind speed
and high relative humidity, which, combined with the relatively high concentration of
PM, would promote the hydrolysis of $N_2O_5$ (Riemer, 2003).

$N_2O_5$ concentrations were calculated by using steady-state expressions (Osthoff et

al., 2006;Wang et al., 2014;Wen et al., 2015), and the result was shown in Fig. 9. The
calculated $N_2O_5$ exhibited a much higher concentration during the night of 30
November compared to the days before and after. Particulate nitrate formed from $N_2O_5$
hydrolysis was then computed during the 12-hour period. Nitrate concentration at 18:00
of 30 November, 2015 (20 $\mu g\ m^{-3}$) was selected as the initial value, and 31 $\mu g\ m^{-3}$ of
particulate nitrate was produced in the following 12 hours, suggesting that approximate
80% of increased particulate nitrate can be attributed to the hydrolysis of $N_2O_5$ in this
case.

To further understand the contribution of $N_2O_5$ hydrolysis, sampling days with

daily-averaged nitrate concentration exceeding the mean plus twice the standard
deviation were selected as the nitrate episode days. In total, 18 episode-days were
selected during the 2-year measurement, with 16 days in winter and the other 2 days in
biomass burning season. In Fig. 10, we presented the averaged diurnal pattern of
particulate nitrate and its related parameters on the 18 selected episode and pre-episode
days. For the episode days, particulate nitrate revealed a similar diurnal pattern as that
of the whole winter (Fig. 6). Nitrate maintained a high concentration during the whole
day with a small peak around 10:00 in the morning. However, for the pre-episode days,
a clear build-up of nitrate can be observed, especially during the nighttime from 17:00
of the pre-episode days to 1:00 of the episode days (as marked in Fig. 10). The average
increment of ammonium nitrate has exceeded 24 $\mu g\ m^{-3}$ during this 9-hour period of
the pre-episode nighttime. The total WIS also increased during this period, which was



mostly attributed to ammonium nitrate (almost 90%) and resulted in an evident increase
of the ratio of nitrate to total WSI. Compared to nitrate, black carbon, a tracer of primary
emissions, showed little change during the pre-episode day. The retroplume showed in
Fig. S4 suggested that the air masses arrived at the SORPES station on the pre-episode
and episode days were almost the same. These results suggest that secondary formation
other than accumulation was the major contributor to the observed increase of
particulate nitrate.
Since the observed nitrate formation mostly occurred during the nighttime of pre-
episode days when the photochemical production of nitric acid would be largely
suppressed, $N_2O_5$ hydrolysis is thus believed to be the major contributor. As showed in
Fig. 10, compared to those during episode days, $NO_2$ concentration was comparable,
but $O_3$ concentration was higher during pre-episode days. This resulted in a higher
production rate of $N_2O_5$ proxy ($NO_2 \times O_3$) in pre-episode days, and favored formation of
nitrate from the hydrolysis of $N_2O_5$. We further calculated the contribution of $N_2O_5$
hydrolysis to nitrate formation during the periods from 17:00 to 23:00 of each pre-
episode day (excluding 2 windy days). A good correlation (R=0.8) was observed
between the calculated nitrate and observed nitrate (Fig. 11), with the slope of 0.8,
indicating most of the observed nitrate formation during nitrate episodes were attributed
to the hydrolysis of $N_2O_5$.
**4.   Summary and Conclusion**
Online measurements of fine particulate nitrate along with trace gases and $PM_{2.5}$ mass
concentrations were conducted for two years from March 2014 to February 2016 using
a MARGA at SORPES station, a rural receptor site in the Yangtze River Delta, eastern
China. Hourly nitrate concentration varied from 0.5 μg m$^{-3}$ to 92.6 μg m$^{-3}$, with an
averaged value of 15.8 μg m$^{-3}$, which was generally higher than the measurement at the
sites in YRD and PRD, but lower than that at the sites in North China Plain. The
contribution of nitrate to total WSI increased from 25% with WSI concentration lower
than 20 μg m$^{-3}$, to 40% when WSI was higher than 140 μg m$^{-3}$, suggesting a major





driver of nitrate to the aerosol pollution in YRD. $NH_3$ is enough to neutralize the acidic
compounds of aerosol, ammonium nitrate was thus the predominate form of the
observed particulate nitrate.   A clear seasonal variation of nitrate was observed with
peak value in January and December and lowest value in August and September.
Biomass burning plumes contributed the nitrate concentration evidently and resulted in
a secondary peak during June. In summer, thermodynamic equilibrium was the major
factor influencing the diurnal variation of nitrate, and resulted in a much lower
concentration at noontime. Nevertheless, the observed nitrate at noontime was
considerably higher than the value predicted by ISORROPIA II model, indicating a
strong production of nitrate by the photo-oxidation of $NO_2$. Air masses from YRD and
biomass burning region were corresponded to the high nitrate concentrations during
summer. In winter, the diurnal variation of nitrate was weak. Regional transport from
North China Plain contributed largely to the observed high nitrate concentrations.

Nitrate episodes, defined as daily-averaged concentration exceeding the mean

value plus twice the standard deviation, were further investigated to understand the
chemical processes towards forming particulate nitrate and their contribution to the
pollution episodes. A clear build-up of nitrate can be observed during the pre-episode
night, and dominated the increase of total WSI. $N_2O_5$ hydrolysis was demonstrated to
contribute 80% of the observed nitrate formation, suggesting its critical role in an
aerosol pollution episode. In view of the significant emission of NO, which is the main
sink of $N_2O_5$ during night, stronger production of $N_2O_5$ is expected at the upper
boundary layer, e.g. residual layer, and contribute to the nitrate formation in the entire
boundary layer. In summary, our study provides evidence that particulate nitrate
especially that formed from $N_2O_5$ hydrolysis is a crucial contributor to the aerosol
pollution episodes in eastern China.
*Data availability*. The GDAS data used in the HYSPLIT calculation can be acquired
from ftp://arlftp.arlhq.noaa.gov/pub/archives/gdas1. Measurement data at SORPES,



including aerosol data and relevant trace gases as well as meteorological data, are
available upon request from the corresponding author before the SORPES database is
opened publicly.
*Acknowledgements.* The research was supported by National Key Research &
Development Program of China (2016YFC0200500) and National Science Foundation
of China (41725020, 91544231, 91644218, 41422504, 91744311, 41675145).

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





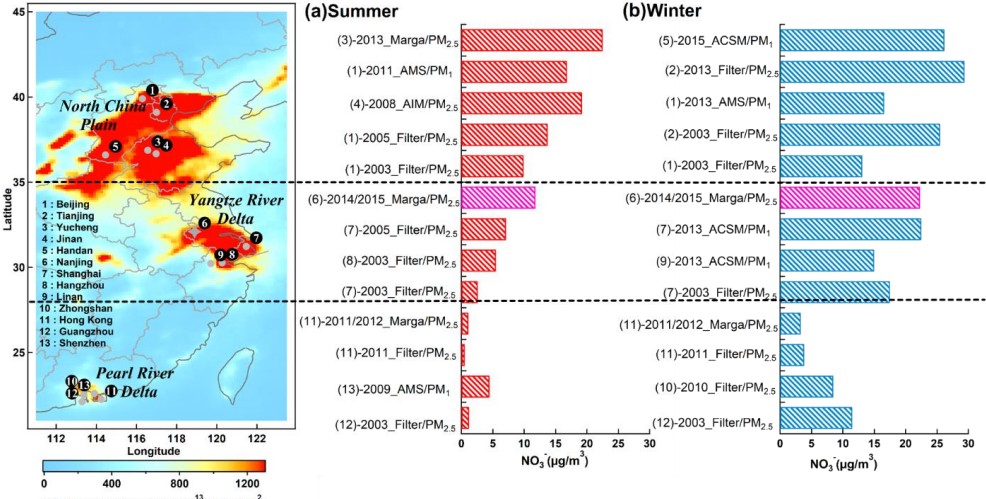

**Figure 1** Average mass concentrations of particulate nitrate at different sampling sites in **(a)** summer and **(b)** winter. The left panel shows the map color-coded by 2-years (2014-2015) averaged tropospheric NO₂ from OMI satellite (http://www.temis.nl/airpollution/no2.html). The pink bars are for this study.



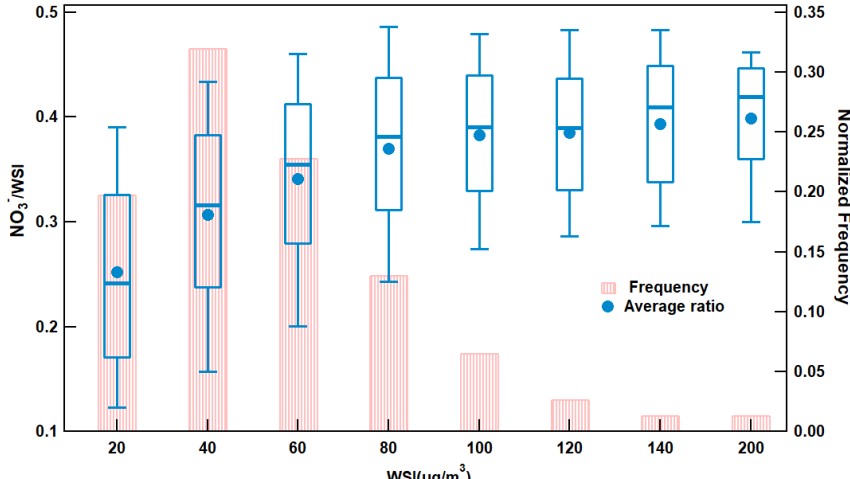

**Figure 2** Average proportion of nitrate and normalized frequency of occurrence at different mass concentration bins of water soluble ions at SORPES. For the ratio, box boundaries represent the interquartile range, bars represent 5%-95% percentile range, and horizontal lines represent the median value.





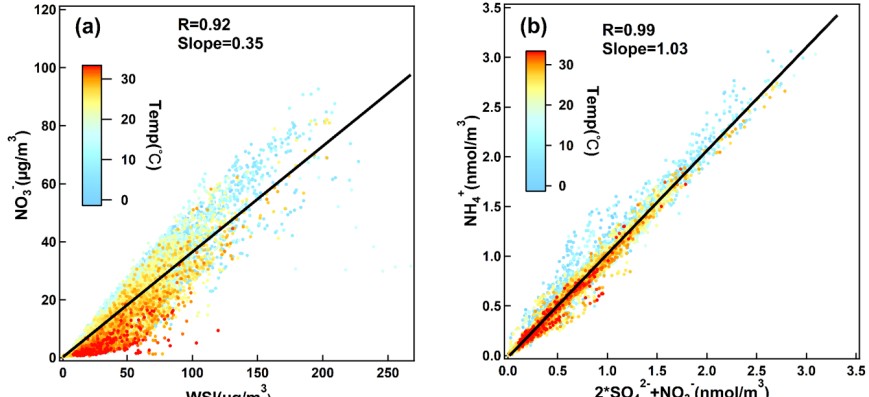

**Figure 3** Scatter plots of **(a)** nitrate vs. total WSI color coded by air temperature, **(b)** molar concentrations of ammonium with nitrate molar concentrations plus two times of sulfate molar concentrations.





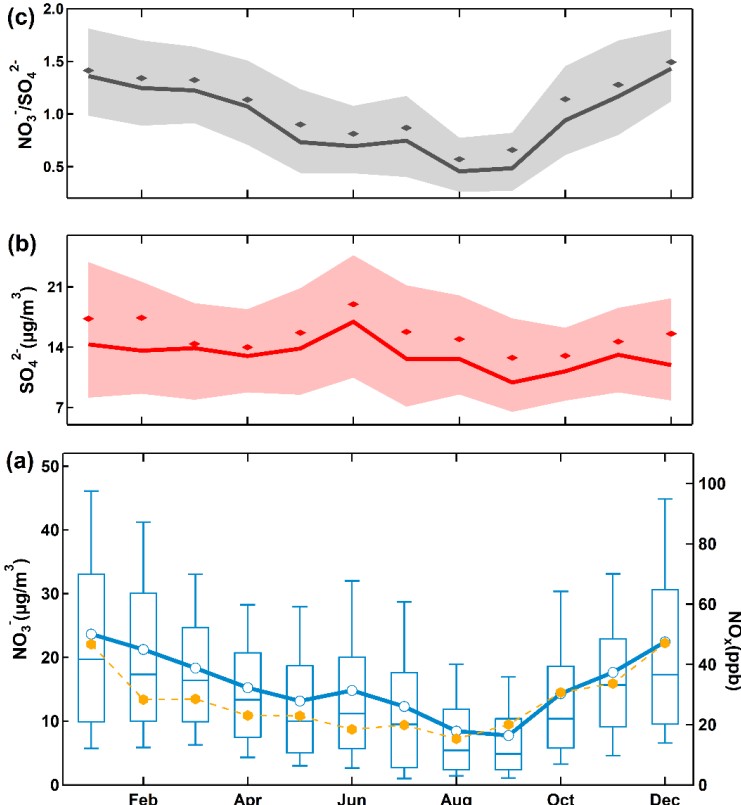

**Figure 4** Monthly averaged nitrate (blue), sulfate (red), $NO_x$ (orange) mass concentrations and nitrate to sulfate ratio (grey) measured at SORPES station during March 2014 to February 2016. For nitrate to sulfate ratio (a) and sulfate (b), bold solid lines are the median values, shade areas represent percentiles of 75% and 25%, and diamonds represent the mean values. For nitrate (c), box boundaries represent the interquartile range, bars represent 5%-95% percentile range, horizontal lines represent the median value, and crosses represent mean values.





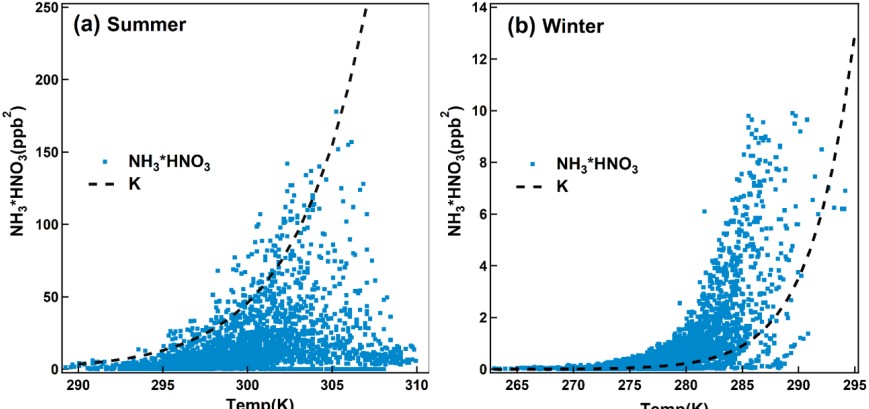

**Figure 5** Scatter plot of product of $[NH_3(g)]*[HNO_3(g)]$ with temperature together with the K parameter calculated by temperature for **(a)** summer and **(b)** winter.





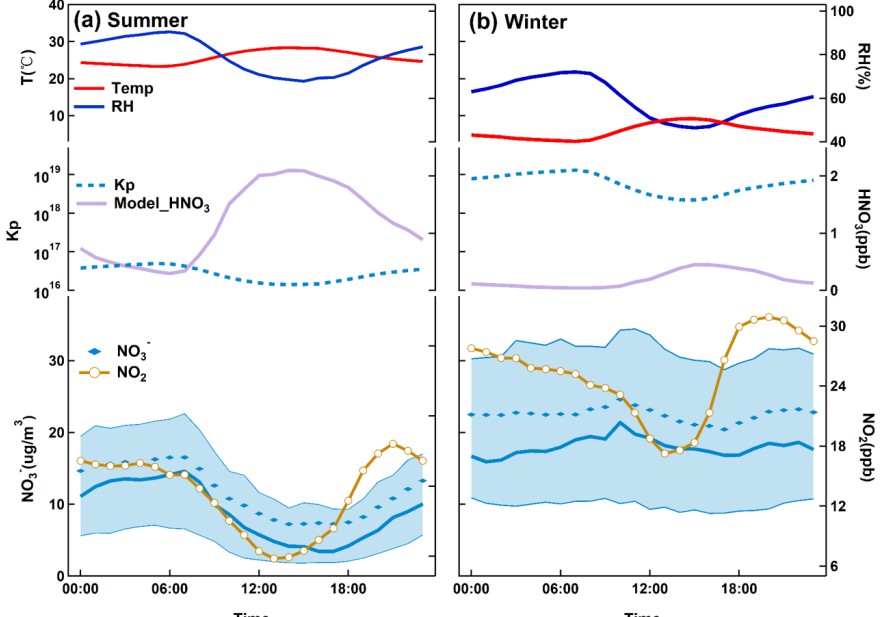

**Figure 6** Diurnal variation of particulate nitrate in **(a)** summer and **(b)** winter. For nitrate, bold solid lines are the median values, shaded areas represent percentiles of 75% and 25% and diamonds represent mean values. Diurnal averages of $NO_2$ and modeled nitric acid mass concentrations are also provided with temperature and RH.



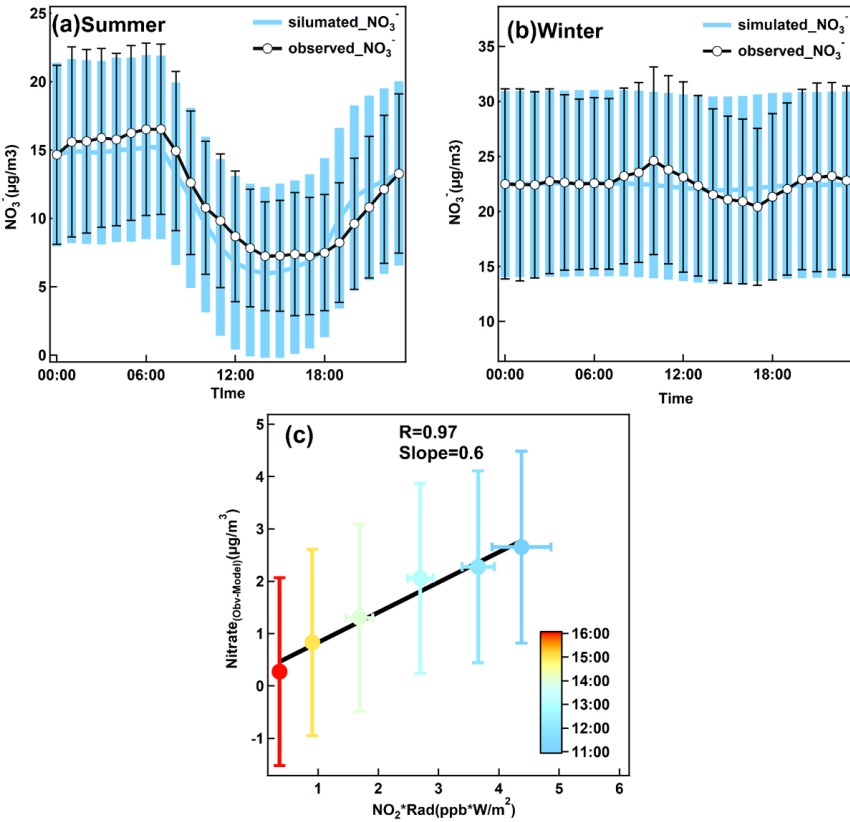

**Figure 7** Modeled nitrate diurnal variations in **(a)** summer and **(b)** winter, together with the observed nitrate concentrations. Error bars provided are the standard deviation of the mean at each hourly interval. **(c)** Scatter plot of the difference between model and observed nitrate average mass concentrations with the product of $NO_2$ and radiation color coded by the hour of day for the samples.



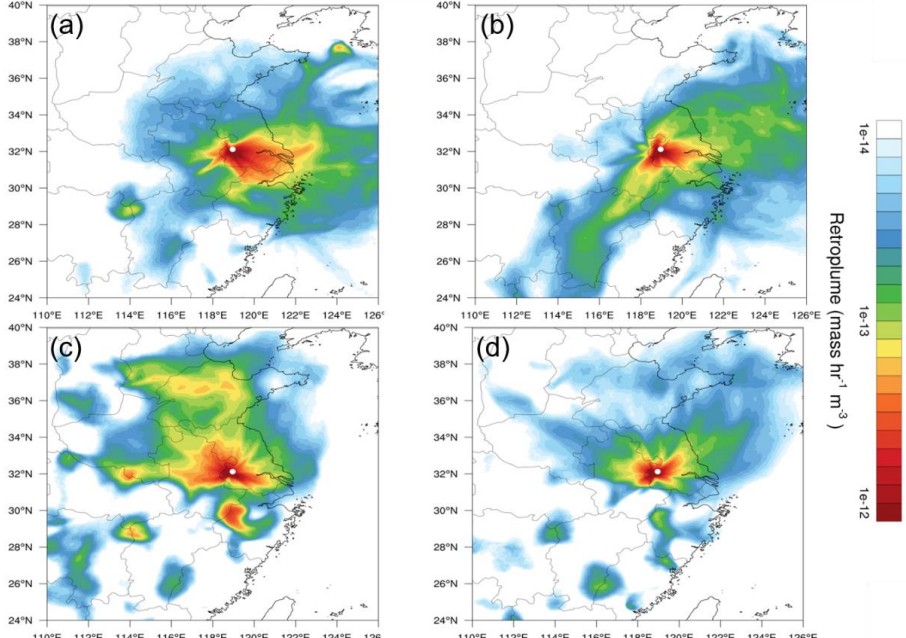

**Figure 8** The averaged retroplumes (i.e., 100 m footprint) of the selected events: **(a)** Top 25% nitrate concentrations in summer, **(b)** Bottom 25% nitrate concentrations in summer, **(c)** Top 25% nitrate concentrations in winter, and **(d)** Bottom 25% nitrate concentrations in winter.



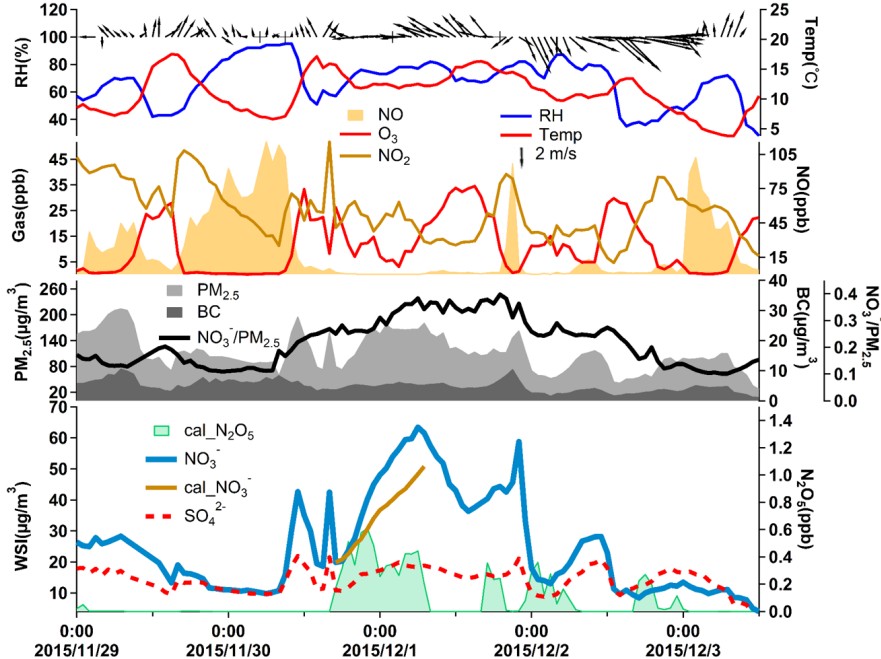

**Figure 9** Time series of meteorological data and the concentrations of trace gases related to nitrate formation during 29 November to 3 December, 2015. Cal_NO$_3^-$ represents the nitrate concentrations calculated from the hydrolysis of N$_2$O$_5$.


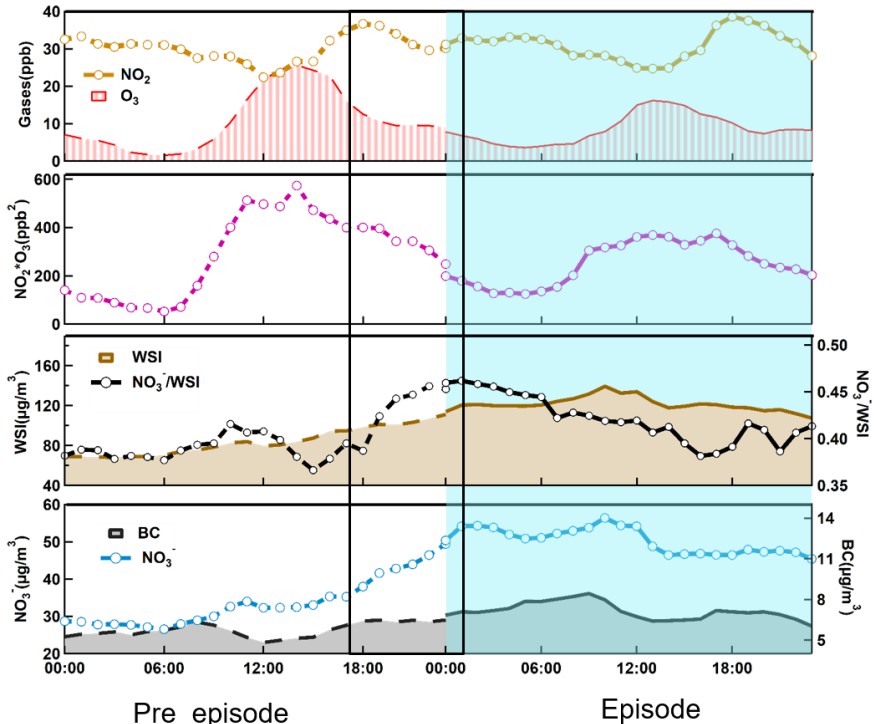

**Figure 10** Diurnal variations of particulate nitrate, black carbon, the total water soluble ions, nitrate to WSI ratio, product of $NO_2$ and $O_3$, $NO_2$, and $O_3$ averaged for nitrate episode days with exceedances of one mean plus two standard deviations. The left side shows the pre-episode days and the right side shows the episode days during the winter of entire two years period. The solid line box corresponds to the rapid growth of nitrate at night.





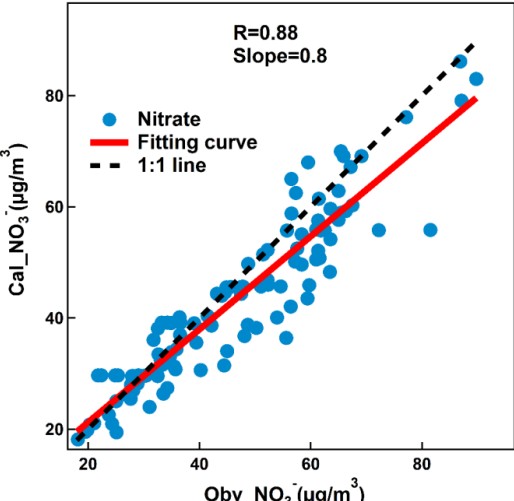

**Figure 11** Scatter plot of calculated nitrate concentrations and observed nitrate concentrations from 17:00 to 23:00 of each episode.





Table 1 major gas phase and heterogeneous reactions involved $NO_3$ and $N_2O_5$

| Reaction | Rate constant |
|---|---|
| $NO_2+O_3\rightarrow NO_3+O_2$ | $k_1$ |
| $NO_3+NO_2\leftrightarrow N_2O_5$ | $k_{eq}$ |
| $NO_3+NO\leftrightarrow NO_2+NO_2$ | $k_3$ |
| $NO_3\rightarrow NO+O_2$ | $j_4$ |
| $NO_3\rightarrow NO+O_2$ | $j_5$ |
| $NO_3 \xrightarrow{voc} products$ | $k_6=\sum (k_{voci}\cdot[voc]_i)$ |
| $NO_3 \xrightarrow{Heterogeneous} products$ | $k_7=0.25\cdot i\cdot C_{NO_3}\cdot\gamma_{NO_3}\cdot S_{aerosol}$ |
| $N_2O_5 \xrightarrow{Heterogeneous} products$ | $k_8=0.25\cdot C_{N_2O_5}\cdot\gamma_{N_2O_5}\cdot S_{aerosol}$ |
| $N_2O_5 \xrightarrow{Homogeneous} 2HNO_3$ | $k_9=k_I\cdot[H_2O]+k_{II}\cdot[H_2O]^2$ |