# Peer review of "Two years online measurement of fine particulate nitrate in western"

_Atmospheric Chemistry and Physics, 2018_

## Referee Comment (RC1) · Anonymous Referee #1 · 9 Aug 2018

This paper reports aerosol composition, its seasonal cycle, its correlation with other trace gases, and an analysis of chemical mechanisms responsible for particulate nitrate formation from a site in the Yangtze River Delta (YRD) of China during two years of continuous measurements at hourly time resolution. The data set and analysis appear to be unique, and their presentation represents a new contribution that will be of interest to the readership of ACP. The paper will add to the growing literature on the characteristics of nitrate aerosol in China. I recommend publication following attention to the comments and technical corrections below.

[Figure]

Minor comments

Line 69: The daytime concentration of N2O5 cannot always be neglected. In some cases, there is evidence that it leads to relatively rapid soluble nitrate production. See Figure 14 in:

Brown, S.S., W.P. Dubé, Y.J. Tham, Q. Zha, L. Xue, S. Poon, Z. Wang, D.R. Blake, W. Tsui, D.D. Parrish, and T. Wang, Nighttime chemistry at a high altitude site above Hong Kong. Journal of Geophysical Research: Atmospheres, 2016. 121(5): p. 2457-2475.

Line 71: The direct water vapor reaction is much slower than heterogeneous uptake and can generally be neglected. The last line in Table 1 shows this reaction using the Wahner parameterization. This parameterization has been shown to be inconsistent with field measurements of N2O5.

Crowley, J.N., J. Thieser, M.J. Tang, G. Schuster, H. Bozem, Z.H. Beygi, H. Fischer, J.M. Diesch, F. Drewnick, S. Borrmann, W. Song, N. Yassaa, J. Williams, D. P$\sqrt{\partial}$hler, U. Platt, and J. Lelieveld, Variable lifetimes and loss mechanisms for NO3 and N2O5 during the DOMINO campaign: contrasts between marine, urban and continental air. Atmos. Chem. Phys., 2011. 11(21): p. 10853-10870.

Brown, S.S., W.P. Dubé, H. Fuchs, T.B. Ryerson, A.G. Wollny, C.A. Brock, R. Bahreini, A.M. Middlebrook, J.A. Neuman, E. Atlas, M. Trainer, F.C. Fehsenfeld, and A.R. Ravishankara, Reactive uptake coefficients for N2O5 determined from aircraft measurements during TexAQS 2006; Comparison to current model parameterizations. J. Geophys. Res., 2009. 114: p. D00F10.

Line 195-197: There is not much basis for the assumption of equal NO3 and N2O5 loss rate constants. It would be useful for the authors to also give the average ambient NO2 level, and the associated average ratio of N2O5 to NO3 calculated from equilibrium. If this ratio is large, then one could argue (with some basis) that N2O5 reactions are likely to be more important than NO3 reactions. Also, what does the symbol "i" represent in

the NO3 uptake expression in Table 1?

Lines 230-233: The trends in nitrate are not evident in Figure 1. To which data do the statements about trends refer?

Line 251: Replace "around 0 C" with a statement of upper and lower bounds, i.e., -5 to +5 C or whatever range defines this percentage of nitrate.

Line 257: The equation in the text line does not make sense. Authors should check for accuracy. Furthermore, it is rare that excess ammonium is observed in the particle phase. Is this what the authors mean to say?

Line 258-262: The seasonal differences referred to here are not apparent in the way the data are presented in Figure 3b. Are the authors invoking Ca, K and Cl to explain the variation of the darker and warmer colors with respect to the fit line? If so, the writing is not clear. If not, then the data for C, K and Cl should be shown.

Line 270: The bimodal pattern is not obvious in sulfate. There does not appear to be a peak in January. If the data were displayed with the y-axis from zero, there would seem to be very little seasonal variation in sulfate. This observation is itself in contrast to other polluted regions (Europe, US), which show a strong summertime maximum in sulfate.

Lines 287-289: Writing is unclear. Is the NOx decrease from Jan – Feb caused by a festival? It would seem more likely to be caused by meteorology / BL depth / transport, etc., but the cause and effect with the festival is implied but not stated. The attribution to factors other than local emissions is therefore not clearly made. Grammar also needs correcting: "It might suggest" should be replaced by "The observations might suggest". Even with the grammar correction, the case for the attribution here is not clear.

Line 290-299: The results of the equilibrium calculation do not make sense. HNO3 is a calculated quantity from the equilibrium. If so, then the points should all lie either exactly on the lines or below it, but not above, since HNO3 above the line would be

[Figure]

calculated to be in the aerosol phase. How was the calculation of HNO3 done, and how does it lead to points that are not in equilibrium under conditions where the aerosol is favored? Also, the plots would be better displayed with the y-axis on a log scale to better illustrate the behavior at low temperature, especially in winter.

Line 327: Brown and Dube 2007 is not the best reference here with respect to particulate nitrate. Baasandorj et al. 2017 is a good reference, however.

Baasandorj, M., S.W. Hoch, R. Bares, J.C. Lin, S.S. Brown, D.B. Millet, R. Martin, K. Kelly, K.J. Zarzana, C.D. Whiteman, W.P. Dubé, G. Tonnesen, I.C. Jaramillo, and J. Sohl, Coupling between Chemical and Meteorological Processes under Persistent Cold-Air Pool Conditions: Evolution of Wintertime PM2.5 Pollution Events and N2O5 Observations in Utah's Salt Lake Valley. Environmental Science & Technology, 2017. 51(11): p. 5941-5950.

Line 354-355: The influence of thermodynamics is not smaller in winter compared to summer. Perhaps the authors mean that it has a smaller influence on the diurnal cycle?

Line 364: Does "percent" mean "percentile"? The text does not make the choice of 25th percentile clear, nor that the selection is for top and bottom percentages. The figure 8 caption is clear. Text should read more like the figure caption.

Line 371-372: The retroplume in Figure S3 does not overlap with the biomass burning region. Does this imply that the region with high biomass burning gives rise to lower nitrate concentrations? What is the overlap of the lower 25th percentages with the biomass burning regions?

Line 432-433: The product of NO2*O3 is a proxy for the N2O5 production rate, but this could be calculated quantitatively in units such as molecules cm-3 s-1 or ppbv hr-1 quite easily by also multiplying by the NO2 + O3 rate constant. This would be more intuitive in Figure 10.

Technical corrections

Line 49: the Chinese government

Line 68: the N2O5 concentration

Line 79 (and 89): do the authors mean "undenuded" rather than "undenude" ?

Line 85: suggest to replace "super" with either "rather" or "extremely"

Line 107: Please specify which Zhang reference (a, b or c)

Line 113: "of" in place of "on"

Line 239: "ranges" instead of "range"

Line 248: Suggest to replace "They overall overall correlated to each other with correlation coefficient . . ." with "The correlation coefficient was . . ."

Line 256: replace "contrasts with" with "in contrast with"

Line 280: eliminate the word "commendably"

Line 291: "calculate" rather than "calculated"

Line 312: replace "prefer to evaporate and dilute the particulate nitrate" with "lead to evaporation and dilution of the particulate nitrate."

Line 316: "The equilibrium constant"

Line 319: Suggest replacing "was highly correlated to" with "showed the same diurnal pattern as"

Line 321: Replace "considerable" with "moderate" and eliminate the word "appeared"

Line 335: replace "were showed" with "are shown"

Line 345: "neglected" in place of "ignored"

Line 349: "product of NO2" rather than "production of NO2". Also insert "the" before "production rate of nitric acid"

[Figure]

Line 367: "be associated with" rather than "accompany with"

Line 399: the steady state approximation

Line 405: "approximately" in place of "approximate"

Line 419: remove the word "has"

Line 450: , and ammonium nitrate

Line 453: contributed to the nitrate

Line 457: the ISORROPIA II model

Line 459: the biomass burning regions

Line 459: Replace "corresponded to" with "associated with"

Line 460: the North China Plain

Line 466: replace "and" with "which"

Line 470: use the phrase "and this residual layer nitrate will contribute"
* * *

---

## Referee Comment (RC2) · Anonymous Referee #2 · 7 Sep 2018

The authors present two-years measurements of fine nitrate aerosol at a rural site in the Yangtze River Delta, China. The data are analyzed to illustrate the seasonal and diurnal variations of particulate nitrate and its formation pathways. It was found that photochemical formation of nitric acid and its thermodynamic equilibrium with NO3- play a dominant role in summer, whilst hydrolysis of N2O5 dominates in winter. Overall, this study provides valuable observational data and useful insights into the chemical behaviors of nitrate aerosol in the polluted atmospheres of China. Thus this manuscript can be accepted for publication after the following comments are properly addressed.

Specific comments:

The major concern is on the steady state calculation of N2O5 and its contribution to the NO3- formation. First, is the steady state assumption valid in this study, especially for the cold conditions in winter? The authors need estimate the chemical lifetimes of N2O5 for the selected cases and examine if the air masses were in steady state for N2O5? Some parameters (e.g., uptake coefficient of N2O5 onto particles) are highly uncertain, which may introduce large uncertainty to the calculation. The chemical loss of NO3 radical via reaction with VOCs is also highly variable and depends on the abundances and chemical speciation of VOCs, especially biogenic VOCs. The authors may conduct more calculations with varying levels of uptake coefficients and reaction rates of NO3+VOCs, to examine the sensitivity of the major conclusions to these assumptions.

Section 3.1: although this manuscript focused on fine particulate nitrate, it should be useful to document the overall measurement results of other related species, such as sulfate, PM2.5, NOx, O3 and NH3. Besides the ratio of nitrate to water-soluble ions, it is also very useful to show the mass ratio of nitrate to PM2.5.

Pg 3, Lines 64-66: the following recent observational studies of N2O5 in China should be acknowledged here.

Wang T. et al., Observations of nitryl chloride and modeling its source and effect on ozone in the planetary boundary layer of southern China, J. Geophys. Res., 121, 2476-2489, 2016.

Wang X. et al., Observations of N2O5 and ClNO2 at a polluted urban surface site in North China: High N2O5 uptake coefficients and low ClNO2 product yields. Atmospheric Environment, 156, 125-134, 2017.

Pg 9, Lines 230-233: it should be noted here that these trends were derived from various observations obtained from different sites in the specific regions, other than

from long-term observations at the same site.

Pg 23, Lines 666-669: cite the final ACP paper instead.

Wen, L. et al., Summertime fine particulate nitrate pollution in the North China Plain: increasing trends, formation mechanisms, and implications for control policy, Atmospheric Chemistry and Physics, 18, 11261-11275, DOI: 10.5194/acp-18-11261-2018, 2018.

Figure 1: the above reference (Wen et al., 2018) has reported very recent observations of fine particulate nitrate at three different sites (urban, rural and mountain sites) in the North China Plain. It would be useful to include these recent data in Figure 1 for comparison.

Figure 4: is the nitrate/sulfate ratio mass-based or molar-based? The molar ratio of nitrate to sulfate should be better here.

Figure 9: please provide a legend for the wind vectors.

Table 1: it should be helpful to provide the exact values of these rate constants used in this study.
* * *

---

## Author Comment (AC1) · 19 Oct 2018

**Referee #2**

The authors present two-years measurements of fine nitrate aerosol at a rural site in the Yangtze River Delta, China. The data are analyzed to illustrate the seasonal and diurnal variations of particulate nitrate and its formation pathways. It was found that photochemical formation of nitric acid and its thermodynamic equilibrium with NO3- play a dominant role in summer, whilst hydrolysis of N2O5 dominates in winter. Overall, this study provides valuable observational data and useful insights into the chemical behaviors of nitrate aerosol in the polluted atmospheres of China. Thus this manuscript can be accepted for publication after the following comments are properly addressed.

Specific comments:

The major concern is on the steady state calculation of N2O5 and its contribution to the NO3- formation. First, is the steady state assumption valid in this study, especially for the cold conditions in winter? The authors need estimate the chemical lifetimes of N2O5 for the selected cases and examine if the air masses were in steady state for N2O5? Some parameters (e.g., uptake coefficient of N2O5 onto particles) are highly uncertain, which may introduce large uncertainty to the calculation. The chemical loss of NO3 radical via reaction with VOCs is also highly variable and depends on the abundances and chemical speciation of VOCs, especially biogenic VOCs. The authors may conduct more calculations with varying levels of uptake coefficients and reaction rates of NO3+VOCs, to examine the sensitivity of the major conclusions to these assumptions.

Response: Thanks, we agree with the referee's comment.

We calculated the chemical lifetimes of $N_2O_5$ using the method described in (Brown et al., 2006 and Brown et al., 2016). The average chemical lifetimes of N2O5 for the selected cases is about 0.1(0.006-0.3) hour, mostly less than 10 minutes (>75%). Air mass of the selected cases were relative stable with low wind speed and consistent wind direction. Therefore, we believed the steady state method can be used in this study.

The uptake coefficient of $N_2O_5$ and the chemical speciation of VOCs can really cause uncertainty. The details of the calculation and the uncertainty to the result will be discussed in support information. We will add the result of uncertainty calculation in the revised manuscript.

[Figure]

Fig. S5 The uncertainty of the calculation of nitrate from hydrolysis of N2O5 with varying levels of uptake coefficients and reaction with different VOCs. It should be noted that the figure is color-coded by the ratio of nitrate produced by hydrolysis of N2O5 with different levels of uptake coefficients and VOCs to the nitrate produced by hydrolysis of N2O5 with the parameter used in the paper section 2.4. The X axis presents the different VOC percentiles.

*Brown, S. S., Ryerson, T. B., Wollny, A. G., Brock, C. A., Peltier, R., Sullivan, A. P., Weber, R. J., Dube, W. P., Trainer, M., Meagher, J. F., Fehsenfeld, F. C., and Ravishankara, A. R.: Variability in nocturnal nitrogen oxide processing and its role in regional air quality, Science, 311, 67-70, 2006.*

*Brown, S. S., Dube, W. P., Tham, Y. J., Zha, Q. Z., Xue, L. K., Poon, S., Wang, Z., Blake, D. R., Tsui, W., Parrish, D. D., and Wang, T.: Nighttime chemistry at a high altitude site above Hong Kong, J Geophys Res-Atmos, 121, 2457-2475, 10.1002/2015JD024566, 2016.*

Section 3.1: although this manuscript focused on fine particulate nitrate, it should be useful to document the overall measurement results of other related species, such as sulfate, PM2.5, NOx,

O3 and NH3. Besides the ratio of nitrate to water-soluble ions, it is also very useful to show the mass ratio of nitrate to PM2.5.

Response: Thanks for the comment.

Seasonal sulfate concentrations have been drawn in Fig.4. The overall statistical results of other related species will be added to table S2. We will add table S2 in the revised support information,

Comments from Referees: Pg 3, Lines 64-66: the following recent observational studies of N2O5 in China should be acknowledged here.

Response: Thanks for the comment. The references will be added in the revised manuscript.

Comments from Referees: Pg 9, Lines 230-233: it should be noted here that these trends were derived from various observations obtained from different sites in the specific regions, other than from long-term observations at the same site.

Response: Thanks for the comment. We agree the trend shown in Fig.1 were not guarantee. We will modify the statement in the revised manuscript as follows.

*"Third, an overall increasing trend of particulate nitrate was implied in NCP and YRD in the past decade, especially that during summertime."*
*"It should be noted that these measurements were from various observations obtained from different sites in specific regions other than from long-term observations at the same site. These conclusions we acquired from limited literature records should have considerable uncertainty."*

Comments from Referees: Pg 23, Lines 666-669: cite the final ACP paper instead.

Response: Thanks for your reminder. The reference will be replaced in the revised manuscript.

Comments from Referees: Figure 1: the above reference (Wen et al., 2018) has reported very recent observations of fine particulate nitrate at three different sites (urban, rural and mountain sites) in the North China Plain. It would be useful to include these recent data in Figure 1 for comparison.

Response: Thanks for your suggestion. We will add the information in the Fig.1 in the revised

manuscript.

Comments from Referees: Figure 4: is the nitrate/sulfate ratio mass-based or molar-based? The molar ratio of nitrate to sulfate should be better here.

Response: Thanks for the comment. The original ratio in the Fig.4 is mass-based and we will replace it with the molar-based ratio in the revised manuscript.

Comments from Referees: Figure 9: please provide a legend for the wind vectors.

Response: Thanks for the comment, and will add a legend for the wind vectors.

Comments from Referees: Table 1: it should be helpful to provide the exact values of these rate constants used in this study.

Response: Thanks for your suggestion. We will add them into the Ttable1 in the revised paper.

---

## Author Comment (AC2) · 19 Oct 2018

**Referee #1**

This paper reports aerosol composition, its seasonal cycle, its correlation with other trace gases, and an analysis of chemical mechanisms responsible for particulate nitrate formation from a site in the Yangtze River Delta (YRD) of China during two years of continuous measurements at hourly time resolution. The data set and analysis appear to be unique, and their presentation represents a new contribution that will be of interest to the readership of ACP. The paper will add to the growing literature on the characteristics of nitrate aerosol in China. I recommend publication following attention to the comments and technical corrections below.

Minor comments

Line 69: The daytime concentration of N2O5 cannot always be neglected. In some cases, there is evidence that it leads to relatively rapid soluble nitrate production.

Response: Yes, we agree that N2O5 cannot be always ignored, especially during the polluted or cloudy days. We will modify the description it in the revised manuscript.

*"Due to the rapid photolysis of $NO_3$ radical, the contribution of $N_2O_5$ hydrolysis to nitrate concentration during daytime of sunny day is usually small."*

Line 71: The direct water vapor reaction is much slower than heterogeneous uptake and can generally be neglected. The last line in Table 1 shows this reaction using the Wahner parameterization. This parameterization has been shown to be inconsistent with field measurements of N2O5.

Response: Thanks to the comment. We agree that the direct water vapor reaction can be neglected compared with the heterogeneous uptake of N2O5. We will remove it in the revised manuscript.

Line 195-197: There is not much basis for the assumption of equal NO3 and N2O5 loss rate constants. It would be useful for the authors to also give the average ambient NO2 level, and the associated average ratio of N2O5 to NO3 calculated from equilibrium. If this ratio is large, then one could argue (with some basis) that N2O5 reactions are likely to be more important than NO3 reactions. Also, what does the symbol "i" represent in the NO3 uptake expression in Table 1?

Response: Thanks for the comment. We will add the NO2 level in the revised manuscript.

We did not have the VOCs measurement during the two-year period, but continuously VOCs measurement using PTR-TOF after 2017. The VOCs data we used in the manuscript was the averaged value measured at SORPES site, which is believed to be a reasonable value. In the revised manuscript, we will recalculate the result about N2O5, and evaluate the uncertainty caused by the uptake coefficient of N2O5 and different levels of VOCs, and will add the information in the support information. We will modify the statement in the revised manuscript.

There should be no symbol "i" in that position in Table 1. Thanks for your reminder.

Lines230-233: The trends in nitrate are not evident in Figure 1. To which data do the statements about trends refer?

Response: Thanks for the comment. We agree that the trends shown in Fig. 1 are not evident, and will modify the statement in the revised manuscript as follows. The related data and references are listed in Table S1. We will modify the description it in the revised manuscript.

*"Third, an overall increasing trend of particulate nitrate was implied in NCP and YRD in the past decade, especially that during summertime"*

*"It should be noted that these measurements were from various observations obtained from different sites in specific regions other than from long-term observations at the same site. These conclusions we acquired from limited literature records should have considerable uncertainty."*

Line 251: Replace "around 0 C" with a statement of upper and lower bounds, i.e., -5 to +5 C or whatever range defines this percentage of nitrate.

Response: Thanks for the comment. We will modify it into the revised manuscript.

Line 257: The equation in the text line does not make sense. Authors should check for accuracy. Furthermore, it is rare that excess ammonium is observed in the particle phase. Is this what the authors mean to say?

Response: Thanks for the comment. Here, excess ammonium is defined as the amount of ammonium in excess of that required for satisfying $[NH_4^+]/[SO_4^{2-}] = 1.5$. The reference is below. If there is not

enough ammonia in the atmosphere, the ammonia tends to react with sulfuric acid and form ammonium hydrogen sulfate first. Then the possibility of ammonia react with nitric acid or ammonium hydrogen sulfate is almost the same. However, here we wanted to express the difference between two sites. The expression is not very proper and we will correct it in the revised manuscript.

*Griffith, S. M., Huang, X. H. H., Louie, P. K. K., and Yu, J. Z.: Characterizing the thermodynamic and chemical composition factors controlling PM2.5 nitrate: Insights gained from two years of online measurements in Hong Kong, Atmospheric Environment, 122, 864-875, 10.1016/j.atmosenv.2015.02.009, 2015.*

Line 258-262: The seasonal differences referred to here are not apparent in the way the data are presented in Figure 3b. Are the authors invoking Ca, K and Cl to explain the variation of the darker and warmer colors with respect to the fit line? If so, the writing is not clear. If not, then the data for C, K and Cl should be shown.

Response: Thanks for the comment. We did not invoke Ca, K and Cl in Fig.3, but in the following figure. High concentrations of Cl can be observed at lower temperature condition. The data points would be below the regression line, when the concentrations of Ca and K were high for some special process in early summer such as dust and biomass burning. We will modify the statement in the revised manuscript.

*"In spring and early summer, a fraction of the particulate nitrate is present in the forms of $Ca(NO_3)_2$ and $KNO_3$, which is the explanation of the points below the regression line in Fig. 3b; while in winter, considerable chloride would consume some ammonia to form $NH_4Cl$ (Hu et al., 2017), resulting in the points below the regression line especially in winter."*

[Figure]

Line 270: The bimodal pattern is not obvious in sulfate. There does not appear to be a peak in January. If the data were displayed with the y-axis from zero, there would seem to be very little seasonal variation in sulfate. This observation is itself in contrast to other polluted regions (Europe, US), which show a strong summertime maximum in sulfate.

Response: Thanks for the comment. We agree that the peak of sulfate in January is not evident. In China, there are usually more $SO_2$ emissions during heating season (winter), especially in northern China. Sulfate concentrations at our site should be influenced by the air masses form Northern China during winter. However, during summer the photochemical reactions of sulfate is stronger. As a result, the seasonal variation of sulfate concentrations is not evident. We will modify the statement in the revised manuscript.

*"Particulate sulfate exhibits a relatively less pronounced seasonal pattern with a small peak in*

*June."*

Lines 287-289: Writing is unclear. Is the NOx decrease from Jan – Feb caused by a festival? It would seem more likely to be caused by meteorology / BL depth / transport, etc., but the cause and effect with the festival is implied but not stated. The attribution to factors other than local emissions is therefore not clearly made. Grammar also needs correcting: "It might suggest" should be replaced by "The observations might suggest".

Even with the grammar correction, the case for the attribution here is not clear.

Response: Thanks for the comment.

Here, we want to explain the big discrepancy between the NOx and nitrate as shown in Fig.4. The NOx concentrations show a big drop. However, nitrate does not. This suggest that the nitrate we observed in February may be more associated with the regional issue/transport instead of local problem. The festival should be one of the reasons of the NOx decrease. Because during festival, people in college town (our site) usually come back to their hometown. As a result, local emissions will be significantly reduced. We will modify the description in the revised manuscript.

*"The observations might suggest that particulate nitrate was influenced by regional transport but not the local emissions in February."*

Line 290-299: The results of the equilibrium calculation do not make sense. HNO3 is a calculated quantity from the equilibrium. If so, then the points should all lie either exactly on the lines or below it, but not above, since HNO3 above the line would be calculated to be in the aerosol phase. How was the calculation of HNO3 done, and how does it lead to points that are not in equilibrium under conditions where the aerosol is favored? Also, the plots would be better displayed with the y-axis on a log scale to better illustrate the behavior at low temperature, especially in winter.

Response: Thanks for the comment. We will remove the plot of equilibrium calculation in the revised manuscript.

The calculation we deployed in Fig. 5 considered only nitrate, ammonia and temperature. And the parameters of dissociation constant can be varied at different situations (Seinfeld and Pandis, 2006). This could the reason of the discrepancy. We agree with the referee's comment and will remove the

plot and related statement in the revised manuscript.

*Seinfeld, J. H., and Pandis, S. N.: Atmospheric Chemistry and Physics: From Air Pollution to Climate Change, John Wiley & Sons, New York, 2nd edition, 1232 pp., 13: 978-0-471-72018-8 2006.*

Line 327: Brown and Dube 2007 is not the best reference here with respect to particulate nitrate. Baasandorj et al. 2017 is a good reference, however.

Response: Yes, thanks. We will replace Brown by Dube 2007 by Baasandorj et al. 2017 in the revised manuscript.

Line 354-355: The influence of thermodynamics is not smaller in winter compared to summer. Perhaps the authors mean that it has a smaller influence on the diurnal cycle?

Response: Yes, thanks. We mean that the influence of thermodynamics is smaller on the diurnal cycle in winter. We will modify the description in the revised manuscript.

Line 364: Does "percent" mean "percentile"? The text does not make the choice of 25th percentile clear, nor that the selection is for top and bottom percentages. The figure 8 caption is clear. Text should read more like the figure caption.

Response: Yes, it should be replaced by "percentile". Thanks for your reminder. We will correct the expression in the revised manuscript.

Line 371-372: The retroplume in Figure S3 does not overlap with the biomass burning region. Does this imply that the region with high biomass burning gives rise to lower nitrate concentrations? What is the overlap of the lower 25th percentages with the biomass burning regions?

Response: Thanks for the comment. The biomass burning activities occurred mostly from May 25 to June 10 instead of the whole summer (Ding et al., 2013). In Figure S3, the main biomass region is in the west and northwest of our site. In Fig.7 we can see that compared to the hours with bottom 25% nitrate concentrations, more air masses came from west and northwest during the hours with nitrate concentrations of top 25% percentile.

*Ding, A. J., Fu, C. B., Yang, X. Q., Sun, J. N., Petäjä, T., Kerminen, V. M., Wang, T., Xie, Y.,*

*Herrmann, E., Zheng, L. F., Nie, W., Liu, Q., Wei, X. L., and Kulmala, M.: Intense atmospheric pollution modifies weather: a case of mixed biomass burning with fossil fuel combustion pollution in eastern China, Atmos. Chem. Phys., 13, 10545-10554, 10.5194/acp-13-10545-2013, 2013.*

Line 432-433: The product of $NO_2*O_3$ is a proxy for the $N_2O_5$ production rate, but this could be calculated quantitatively in units such as molecules cm-3 s-1 or ppbv hr-1 quite easily by also multiplying by the $NO_2 + O_3$ rate constant. This would be more intuitive in Figure 10.

Response: Yes, it is more intuitive and better. Thanks for the comment. We will modify it in the revised manuscript.

Technical corrections:

Line 49: the Chinese government

Response: Thanks. We will correct it in the revised manuscript.

Line 68: the $N_2O_5$ concentration

Response: Thanks. We will correct it in the revised manuscript.

Line 79 (and 89): do the authors mean "undenuded" rather than "undenude" ?

Response: Thanks. We will correct it in the revised manuscript.

Line 85: suggest to replace "super" with either "rather" or "extremely"

Response: Thanks. We will correct it in the revised manuscript.

Line 107: Please specify which Zhang reference (a, b or c)

Response: Thanks. We will correct it in the revised manuscript.

Line 113: "of" in place of "on"

Response: Thanks. We will correct it in the revised manuscript.

Line 239: "ranges" instead of "range"

Response: Thanks. We will correct it in the revised manuscript.

Line 248: Suggest to replace "They overall overall correlated to each other with correlation coefficient …" with "The correlation coefficient was …"

Response: Thanks. We will correct it in the revised manuscript.

Line 256: replace "contrasts with" with "in contrast with"

Response: Thanks. We will correct it in the revised manuscript.

Line 280: eliminate the word "commendably"

Response: Thanks. We will correct it in the revised manuscript.

Line 291: "calculate" rather than "calculated"

Response: Thanks. We will correct it in the revised manuscript.

Line 312: replace "prefer to evaporate and dilute the particulate nitrate" with "lead to evaporation and dilution of the particulate nitrate."

Response: Thanks. We will correct it in the revised manuscript.

Line 316: "The equilibrium constant"

Response: Thanks. We will correct it in the revised manuscript.

Line 319: Suggest replacing "was highly correlated to" with "showed the same diurnal pattern as"

Response: Thanks. We will correct it in the revised manuscript.

Line 321: Replace "considerable" with "moderate" and eliminate the word "appeared"

Response: Thanks. We will correct it in the revised manuscript.

Line 335: replace "were showed" with "are shown"

Response: Thanks. We will correct it in the revised manuscript.

Line 345: "neglected" in place of "ignored"

Response: Thanks. We will correct it in the revised manuscript.

Line 349: "product of NO2" rather than "production of NO2". Also insert "the" before "production rate of nitric acid"

Response: Thanks. We will correct it in the revised manuscript.

Line 367: "be associated with" rather than "accompany with"

Response: Thanks. We will correct it in the revised manuscript.

Line 399: the steady state approximation

Response: Thanks. We will correct it in the revised manuscript.

Line 405: "approximately" in place of "approximate"

Response: Thanks. We will correct it in the revised manuscript.

Line 419: remove the word "has"

Response: Thanks. We will correct it in the revised manuscript.

Line 450: , and ammonium nitrate

Response: Thanks. We will correct it in the revised manuscript.

Line 453: contributed to the nitrate

Response: Thanks. We will correct it in the revised manuscript.

Line 457: the ISORROPIA II model

Response: Thanks. We will correct it in the revised manuscript.

Line 459: the biomass burning regions

Response: Thanks. We will correct it in the revised manuscript.

Line 459: Replace "corresponded to" with "associated with"

Response: Thanks. We will correct it in the revised manuscript.

Line 460: the North China Plain

Response: Thanks. We will correct it in the revised manuscript.

Line 466: replace "and" with "which"

Response: Thanks. We will correct it in the revised manuscript.

Line 470: use the phrase "and this residual layer nitrate will contribute"

Response: Thanks. We will correct it in the revised manuscript.

---

## Author Response (AR1)

**Referee #1**

This paper reports aerosol composition, its seasonal cycle, its correlation with other trace gases, and an analysis of chemical mechanisms responsible for particulate nitrate formation from a site in the Yangtze River Delta (YRD) of China during two years of continuous measurements at hourly time resolution. The data set and analysis appear to be unique, and their presentation represents a new contribution that will be of interest to the readership of ACP. The paper will add to the growing literature on the characteristics of nitrate aerosol in China. I recommend publication following attention to the comments and technical corrections below.

Minor comments

Line 69: The daytime concentration of $N_2O_5$ cannot always be neglected. In some cases, there is evidence that it leads to relatively rapid soluble nitrate production.

Response: Yes, we agree that $N_2O_5$ cannot be always ignored, especially during the polluted or cloudy days and modified the description in the revised manuscript.

*"Due to the rapid photolysis of $NO_3$ radical, the contribution of $N_2O_5$ hydrolysis to nitrate concentration during daytime of sunny day is usually small."*

Line 71: The direct water vapor reaction is much slower than heterogeneous uptake and can generally be neglected. The last line in Table 1 shows this reaction using the Wahner parameterization. This parameterization has been shown to be inconsistent with field measurements of $N_2O_5$.

Response: Thanks to the comment. We agree that the direct water vapor reaction can be neglected compared with the heterogeneous uptake of $N_2O_5$ and removed it in the revised manuscript.

Line 195-197: There is not much basis for the assumption of equal $NO_3$ and $N_2O_5$ loss rate constants. It would be useful for the authors to also give the average ambient $NO_2$ level, and the associated average ratio of $N_2O_5$ to $NO_3$ calculated from equilibrium. If this ratio is large, then one could argue (with some basis) that $N_2O_5$ reactions are likely to be more important than $NO_3$ reactions. Also, what does the symbol "i" represent in the $NO_3$ uptake expression in Table 1?

Response: Thanks for the comment. We had added the $NO_2$ level in the revised manuscript.

We did not have the VOCs measurement during the two-year period, but continuously VOCs measurement using PTR-TOF after 2017. The VOCs data we used in the manuscript was the averaged nighttime value measured during winter at SORPES site, which is believed to be a reasonable value. In the revised manuscript, we had recalculated the result about $N_2O_5$, and evaluated the uncertainty caused by the uptake coefficient of $N_2O_5$ and different levels of VOCs, and added the information in the support information. We modified the statement in the revised manuscript.

There should be no symbol "i" in that position in Table 1. Thanks for your reminder.

Lines230-233: The trends in nitrate are not evident in Figure 1. To which data do the statements about trends refer?

Response: Thanks for the comment. We agree that the trends shown in Fig. 1 are not evident, and modified the statement in the revised manuscript as follows. The related data and references had been listed in Table S1. We modified the description in the revised manuscript.

*"Third, an overall increase of particulate nitrate was implied in NCP and YRD in the past decade, especially that during summertime"*

*"It should be noted that the dataset cited in Fig. 1 were obtained from different sites with different techniques. Trends inferred from these datasets could suffer from considerable uncertainty."*

Line 251: Replace "around 0 C" with a statement of upper and lower bounds, i.e., -5 to +5 C or whatever range defines this percentage of nitrate.

Response: Thanks for the comment. We modified it in the revised manuscript.

Line 257: The equation in the text line does not make sense. Authors should check for accuracy. Furthermore, it is rare that excess ammonium is observed in the particle phase. Is this what the authors mean to say?

Response: Thanks for the comment. Here, excess ammonium is defined as the amount of ammonium in excess of that required for satisfying $[NH_4^+]/[SO_4^{2-}] = 1.5$. The reference is listed here. If there is

not enough ammonia in the atmosphere, the ammonia tends to react with sulfuric acid and form ammonium hydrogen sulfate first. Then the possibility of ammonia react with nitric acid or ammonium hydrogen sulfate is comparable. However, here we wanted to express the difference between two sites. The modified the related statement in the revised manuscript.

*Griffith, S. M., Huang, X. H. H., Louie, P. K. K., and Yu, J. Z.: Characterizing the thermodynamic and chemical composition factors controlling PM2.5 nitrate: Insights gained from two years of online measurements in Hong Kong, Atmospheric Environment, 122, 864-875, 10.1016/j.atmosenv.2015.02.009, 2015.*

Line 258-262: The seasonal differences referred to here are not apparent in the way the data are presented in Figure 3b. Are the authors invoking Ca, K and Cl to explain the variation of the darker and warmer colors with respect to the fit line? If so, the writing is not clear. If not, then the data for C, K and Cl should be shown.

Response: Thanks for the comment. We did not invoke Ca, K and Cl in Fig.3, but in the following figures. High concentrations of Cl can be observed at lower temperature condition. The data points would be below the regression line, when the concentrations of Ca and K were high for some special process in early summer such as dust and biomass burning. We modified the statement in the revised manuscript.

*"In spring and early summer, a fraction of the particulate nitrate is present as $Ca(NO_3)_2$ and $KNO_3$, which could explain the data points below the regression line in Fig. 3b. In winter, considerable ammonium is existed as $NH_4Cl$ (Hu et al., 2017), resulting in the data points above the regression line."*

[Figure]

Line 270: The bimodal pattern is not obvious in sulfate. There does not appear to be a peak in January. If the data were displayed with the y-axis from zero, there would seem to be very little seasonal variation in sulfate. This observation is itself in contrast to other polluted regions (Europe, US), which show a strong summertime maximum in sulfate.

Response: Thanks for the comment. We agree that the peak of sulfate in January is not evident. In China, there are usually more $SO_2$ emissions during heating season (winter), especially in northern China. Sulfate concentrations at our site should be influenced by the air masses form Northern China during winter. However, during summer the photochemical reactions of sulfate is stronger. As a result, the seasonal variation of sulfate concentrations is not evident. We modified the statement in the revised manuscript.

"*Particulate sulfate exhibits a relatively less pronounced seasonal pattern with a small peak in*

*June."*

Lines 287-289: Writing is unclear. Is the NOx decrease from Jan – Feb caused by a festival? It would seem more likely to be caused by meteorology / BL depth / transport, etc., but the cause and effect with the festival is implied but not stated. The attribution to factors other than local emissions is therefore not clearly made. Grammar also needs correcting: "It might suggest" should be replaced by "The observations might suggest".

Even with the grammar correction, the case for the attribution here is not clear.

Response: Thanks for the comment.

Here, we want to explain the big discrepancy between the $NO_x$ and nitrate as shown in Fig.4. The $NO_x$ concentrations showed a evident drop, but nitrate did not. This may suggest that the nitrate we observed in February may be more associated with the regional issue/transport instead of local problem. The festival should be one of the reasons of the $NO_x$ decrease. Because during festival, people in college town (around our site) usually go back to their hometown. As a result, local emissions will be significantly reduced. We modified the description in the revised manuscript.

*"The observations might suggest that particulate nitrate was influenced by regional transport but not the local emissions in February."*

Line 290-299: The results of the equilibrium calculation do not make sense. $HNO_3$ is a calculated quantity from the equilibrium. If so, then the points should all lie either exactly on the lines or below it, but not above, since $HNO_3$ above the line would be calculated to be in the aerosol phase. How was the calculation of $HNO_3$ done, and how does it lead to points that are not in equilibrium under conditions where the aerosol is favored? Also, the plots would be better displayed with the y-axis on a log scale to better illustrate the behavior at low temperature, especially in winter.

Response: Thanks for the comment. We had removed the plot of equilibrium calculation in the revised manuscript.

The calculation we deployed in Fig. 5 considered only nitrate, ammonia and temperature. And the parameters of dissociation constant can be varied at different situations (Seinfeld and Pandis, 2006). This could the reason of the discrepancy. We agree with the referee's comment and removed the

plot and related statement in the revised manuscript.

*Seinfeld, J. H., and Pandis, S. N.: Atmospheric Chemistry and Physics: From Air Pollution to Climate Change, John Wiley & Sons, New York, 2nd edition, 1232 pp., 13: 978-0-471-72018-8 2006.*

Line 327: Brown and Dube 2007 is not the best reference here with respect to particulate nitrate. Baasandorj et al. 2017 is a good reference, however.

Response: Yes, thanks. We replaced Brown by Dube 2007 by Baasandorj et al. 2017 in the revised manuscript.

Line 354-355: The influence of thermodynamics is not smaller in winter compared to summer. Perhaps the authors mean that it has a smaller influence on the diurnal cycle?

Response: Yes, thanks. We mean that the influence of thermodynamics is smaller on the diurnal cycle in winter and modified the statment in the revised manuscript.

Line 364: Does "percent" mean "percentile"? The text does not make the choice of 25th percentile clear, nor that the selection is for top and bottom percentages. The figure 8 caption is clear. Text should read more like the figure caption.

Response: Yes, it should be replaced by "percentile". Thanks for your reminder and we corrected the expression in the revised manuscript.

Line 371-372: The retroplume in Figure S3 does not overlap with the biomass burning region. Does this imply that the region with high biomass burning gives rise to lower nitrate concentrations? What is the overlap of the lower 25th percentages with the biomass burning regions?

Response: Thanks for the comment. The biomass burning activities occurred mostly from May 25 to June 10 instead of the whole summer (Ding et al., 2013). In Figure S3, the main biomass region is in the west and northwest of our site. In Fig.7 we can see that compared to the hours with bottom 25% nitrate concentrations, more air masses came from west and northwest during the hours with nitrate concentrations of top 25% percentile.

*Ding, A. J., Fu, C. B., Yang, X. Q., Sun, J. N., Petäjä, T., Kerminen, V. M., Wang, T., Xie, Y.,*

*Herrmann, E., Zheng, L. F., Nie, W., Liu, Q., Wei, X. L., and Kulmala, M.: Intense atmospheric pollution modifies weather: a case of mixed biomass burning with fossil fuel combustion pollution in eastern China, Atmos. Chem. Phys., 13, 10545-10554, 10.5194/acp-13-10545-2013, 2013.*

Line 432-433: The product of $NO_2*O_3$ is a proxy for the $N_2O_5$ production rate, but this could be calculated quantitatively in units such as molecules cm-3 s-1 or ppbv hr-1 quite easily by also multiplying by the $NO_2 + O_3$ rate constant. This would be more intuitive in Figure 10.

Response: Yes, it is more intuitive and better. Thanks for the comment and we modified it in the revised manuscript.

Technical corrections:

Line 49: the Chinese government

Response: Thanks. We corrected it in the revised manuscript.

Line 68: the $N_2O_5$ concentration

Response: Thanks. We corrected it in the revised manuscript.

Line 79 (and 89): do the authors mean "undenuded" rather than "undenude" ?

Response: Thanks. We corrected it in the revised manuscript.

Line 85: suggest to replace "super" with either "rather" or "extremely"

Response: Thanks. We corrected it in the revised manuscript.

Line 107: Please specify which Zhang reference (a, b or c)

Response: Thanks. We corrected it in the revised manuscript.

Line 113: "of" in place of "on"

Response: Thanks. We corrected it in the revised manuscript.

Line 239: "ranges" instead of "range"

Response: Thanks. We corrected it in the revised manuscript.

Line 248: Suggest to replace "They overall overall correlated to each other with correlation coefficient …" with "The correlation coefficient was …"

Response: Thanks. We corrected it in the revised manuscript.

Line 256: replace "contrasts with" with "in contrast with"

Response: Thanks. We corrected it in the revised manuscript.

Line 280: eliminate the word "commendably"

Response: Thanks. We corrected it in the revised manuscript.

Line 291: "calculate" rather than "calculated"

Response: Thanks. We corrected it in the revised manuscript.

Line 312: replace "prefer to evaporate and dilute the particulate nitrate" with "lead to evaporation and dilution of the particulate nitrate."

Response: Thanks. We corrected it in the revised manuscript.

Line 316: "The equilibrium constant"

Response: Thanks. We corrected it in the revised manuscript.

Line 319: Suggest replacing "was highly correlated to" with "showed the same diurnal pattern as"

Response: Thanks. We corrected it in the revised manuscript.

Line 321: Replace "considerable" with "moderate" and eliminate the word "appeared"

Response: Thanks. We corrected it in the revised manuscript.

Line 335: replace "were showed" with "are shown"

Response: Thanks. We corrected it in the revised manuscript.

Line 345: "neglected" in place of "ignored"

Response: Thanks. We corrected it in the revised manuscript.

Line 349: "product of $NO_2$" rather than "production of $NO_2$". Also insert "the" before "production rate of nitric acid"

Response: Thanks. We corrected it in the revised manuscript.

Line 367: "be associated with" rather than "accompany with"

Response: Thanks. We corrected it in the revised manuscript.

Line 399: the steady state approximation

Response: Thanks. We corrected it in the revised manuscript.

Line 405: "approximately" in place of "approximate"

Response: Thanks. We corrected it in the revised manuscript.

Line 419: remove the word "has"

Response: Thanks. We corrected it in the revised manuscript.

Line 450: , and ammonium nitrate

Response: Thanks. We corrected it in the revised manuscript.

Line 453: contributed to the nitrate

Response: Thanks. We corrected it in the revised manuscript.

Line 457: the ISORROPIA II model

Response: Thanks. We corrected it in the revised manuscript.

Line 459: the biomass burning regions

Response: Thanks. We corrected it in the revised manuscript.

Line 459: Replace "corresponded to" with "associated with"

Response: Thanks. We corrected it in the revised manuscript.

Line 460: the North China Plain

Response: Thanks. We corrected it in the revised manuscript.

Line 466: replace "and" with "which"

Response: Thanks. We corrected it in the revised manuscript.

Line 470: use the phrase "and this residual layer nitrate will contribute"

Response: Thanks. We corrected it in the revised manuscript.

**Referee #2**

The authors present two-years measurements of fine nitrate aerosol at a rural site in the Yangtze River Delta, China. The data are analyzed to illustrate the seasonal and diurnal variations of particulate nitrate and its formation pathways. It was found that photochemical formation of nitric acid and its thermodynamic equilibrium with $NO_3^-$ play a dominant role in summer, whilst hydrolysis of $N_2O_5$ dominates in winter. Overall, this study provides valuable observational data and useful insights into the chemical behaviors of nitrate aerosol in the polluted atmospheres of China. Thus this manuscript can be accepted for publication after the following comments are properly addressed.

Specific comments:

The major concern is on the steady state calculation of $N_2O_5$ and its contribution to the $NO_3^-$ formation. First, is the steady state assumption valid in this study, especially for the cold conditions in winter? The authors need estimate the chemical lifetimes of N2O5 for the selected cases and examine if the air masses were in steady state for $N_2O_5$? Some parameters (e.g., uptake coefficient of $N_2O_5$ onto particles) are highly uncertain, which may introduce large uncertainty to the calculation. The chemical loss of $NO_3$ radical via reaction with VOCs is also highly variable and depends on the abundances and chemical speciation of VOCs, especially biogenic VOCs. The authors may conduct more calculations with varying levels of uptake coefficients and reaction rates of $NO_3$+VOCs, to examine the sensitivity of the major conclusions to these assumptions.

Response: Thanks, we agree with the referee's comment.

We calculated the chemical lifetimes of $N_2O_5$ using the method described in (Brown et al., 2006 and Brown et al., 2016). The average chemical lifetimes of $N_2O_5$ for the selected cases is about 0.1(0.006-0.3) hour, mostly less than 10 minutes (>75%). Air mass of the selected cases were relative stable with low wind speed and consistent wind direction. Therefore, we believed the steady state method can be used in this study.

The uptake coefficient of $N_2O_5$ and the chemical speciation of VOCs can really cause uncertainty. The details of the calculation and the uncertainty to the result had been discussed in support information and we added the result of uncertainty calculation in the revised manuscript.

[Figure]

Fig. S1 The uncertainty estimation for the calculation of nitrate from $N_2O_5$ hydrolysis with varying uptake coefficients from 0.01 to 0.05 and VOCs concentration. The calculated nitrate production from $N_2O_5$ hydrolysis in section 2.4 was set as the reference value. The figure is color-coded by the ratio of the calculated nitrate with varied uptake coefficients and VOCs concentrations to the reference value. The X axis presents the different nighttime VOCs concentrations measured during the winter of 2017, ranged from 10 percentiles to 90 percentiles.

*Brown, S. S., Ryerson, T. B., Wollny, A. G., Brock, C. A., Peltier, R., Sullivan, A. P., Weber, R. J., Dube, W. P., Trainer, M., Meagher, J. F., Fehsenfeld, F. C., and Ravishankara, A. R.: Variability in nocturnal nitrogen oxide processing and its role in regional air quality, Science, 311, 67-70, 2006.*

*Brown, S. S., Dube, W. P., Tham, Y. J., Zha, Q. Z., Xue, L. K., Poon, S., Wang, Z., Blake, D. R., Tsui, W., Parrish, D. D., and Wang, T.: Nighttime chemistry at a high altitude site above Hong Kong, J Geophys Res-Atmos, 121, 2457-2475, 10.1002/2015JD024566, 2016.*

Section 3.1: although this manuscript focused on fine particulate nitrate, it should be useful to document the overall measurement results of other related species, such as sulfate, PM$_{2.5}$, NO$_x$, O$_3$ and NH$_3$. Besides the ratio of nitrate to water-soluble ions, it is also very useful to show the mass ratio of nitrate to PM$_{2.5}$.

Response: Thanks for the comment.

Seasonal sulfate concentrations have been drawn in Fig.4. We had added Table S2 in the revised support information and the overall statistical results of other related species were added to Table S2.

Comments from Referees: Pg 3, Lines 64-66: the following recent observational studies of N2O5 in China should be acknowledged here.

Response: Thanks for the comment. The references were added in the revised manuscript.

Comments from Referees: Pg 9, Lines 230-233: it should be noted here that these trends were derived from various observations obtained from different sites in the specific regions, other than from long-term observations at the same site.

Response: Thanks for the comment. We agree the trend shown in Fig.1 were not guarantee and we modified the statement in the revised manuscript as follows.

*"Third, an overall increase of particulate nitrate was implied in NCP and YRD in the past decade, especially that during summertime."*

*"It should be noted that the dataset cited in Fig. 1 were obtained from different sites with different techniques. Trends inferred from these datasets could suffer from considerable uncertainty."*

Comments from Referees: Pg 23, Lines 666-669: cite the final ACP paper instead.

Response: Thanks for your reminder and the reference were replaced in the revised manuscript.

Comments from Referees: Figure 1: the above reference (Wen et al., 2018) has reported very recent observations of fine particulate nitrate at three different sites (urban, rural and mountain

sites) in the North China Plain. It would be useful to include these recent data in Figure 1 for comparison.

Response: Thanks for your suggestion and we added the information in the Fig.1 in the revised manuscript.

Comments from Referees: Figure 4: is the nitrate/sulfate ratio mass-based or molar-based? The molar ratio of nitrate to sulfate should be better here.

Response: Thanks for the comment. The original ratio in the Fig.4 is mass-based and we replaced it with the molar-based ratio in the revised manuscript.

Comments from Referees: Figure 9: please provide a legend for the wind vectors.

Response: Thanks for the comment, and we had added a legend for the wind vectors.

Comments from Referees: Table 1: it should be helpful to provide the exact values of these rate constants used in this study.

Response: Thanks for your suggestion and we had added them into the Table1 in the revised paper.

---

## Author Response (AR2)

**Response to editor**

Co-Editor Decision: Publish subject to minor revisions (review by editor) (02 Nov 2018) by Daniel Knopf

Comments to the Author:

Dear Authors,

I am happy to accept your manuscript for publication in Atmospheric Chemistry and Physics after you have addressed the minor comments given below.

With kindest regards,

Daniel Knopf

Minor revisions:

Main text:

l. 225: "Third, an overall increase of particulate nitrate was implied in NCP and YRD in the past decade, especially that during summertime." What does this sentence mean? I assume you have to cite the studies which showed "that particulate nitrate increased in the areas of NCP and YRD during the past decade".

Response: Thanks for your comment. Here, we thought that this increase of nitrate can be implied by the result shown in Fig.1. We also pointed out the uncertainty of our result. We had modified the description in the revised manuscript.

Supplement:

Figure S1 and its figure caption are not clearly presented. Somehow, the actual VOC concentration has to be given, maybe below 10%, 50%, and 90% percentiles and/or in figure caption? Also, the color coded ratio is not well described. It states "by the ratio of the calculated nitrate with varied uptake coefficients". This is difficult to understand. Is it the ratio of nitrate (in which units?) divided by uptake coefficient? Please improve on this description and make it easier for the reader to follow.

Response: Thanks for your reminder. We added the actual VOCs concentrations in table S1. We corrected the description of figure caption of figure S1 in the revised supplement.

Table S1: Please correct citations. For example, Wen, Chen et al., 2018. What does this mean? Is it Wen et al., 2018 and Chen et al., 2018? Also, Pathak and W.S. Wu, 2009. Is it Pathak and Wu, 2009?

Response: Thanks for your reminder. We corrected the citations in the revised supplement.

The reference list is missing in the supplement document

Response: Thanks for your reminder. We added the reference list in the supplement.

[revised manuscript text omitted]

**Supplement**

**1. The uncertainty of the calculation of $N_2O_5$ hydrolysis**

The variation of $N_2O_5$ uptake coefficient and VOCs concentrations are the dominate sources of the uncertainty to calculate $N_2O_5$ hydrolysis. Here, a range of uptake coefficient from 0.01 to 0.05 (Brown et al., 2006; Brown et al., 2016; Wen et al., 2015; Osthoff et al., 2006) with the interval of 0.002 was deployed in the calculation to compare with the result in section 2.4. Since VOCs were not measured at SORPES station during 2014 – 2016, averaged nighttime values those were measured using a PTR-TOF$_{1000}$ in the winter of 2017 were used in the calculation. Four species of isoprene, monoterpene, styrene and phenol, which have higher reaction rates with $NO_3$ radical (Osthoff et al., 2006) and higher concentrations at SORPES station. 10 percentiles, 25 percentiles, median ,75 percentiles and 90 percentiles of night VOCs concentrations in winter with the actual concentrations shown in table S1 have been selected as the input to calculate the uncertainties. The results are shown in Fig. S1. The largest uncertainty, which can be up to 70%, was from the variation of $N_2O_5$ uptake coefficient. The changes of VOCs concentrations can also cause about 30% uncertainty. More experiments are thus recommended to quantify the $N_2O_5$ uptake coefficient and in turn the contribution of $N_2O_5$ hydrolysis to nitrate formation.

[Figure]

**Figure S1** The uncertainty estimation for the calculation of nitrate from $N_2O_5$ hydrolysis. The calculated nitrate production from $N_2O_5$ hydrolysis in section 2.4 was set as the reference value. The figure is color-coded by the ratio of the calculated nitrate to this reference value with varied uptake coefficients and VOCs concentrations. The y axis presents $N_2O_5$ uptake coefficients from 0.01 to 0.05. The x axis presents the nighttime VOCs concentrations measured during the winter of 2017, which were listed in Table S1 and ranged from 10th percentiles to 90th percentiles.

[Figure]

**Figure S2 (a)**Wind rose plot, color-coded by wind speed (m/s). **(b)**Wind rose plot, color-coded by the median nitrate mass concentrations at corresponding wind speed and wind direction.

[Figure]

**Figure S3** Monthly variation of potassium. Bold solid lines are the median values and thin solid lines represent percentiles of 75% and 25%.

[Figure]

**Figure S4** The averaged retroplumes (i.e., 100 m footprint) for the Top 25% nitrate concentrations in summer 2014 together with the fire spots during the same time. Pink points indicate the fire spots.

[Figure]

**Figure S5** Averaged retroplume of air masses at Nanjing for the period of **(a)** pre-episode days and **(b)** episode days of nitrate pollution.

Table S1 The actual concentrations of VOCs different quantiles.

|  | Isoprene (ppb) | Monoterpene (ppb) | Phenol (ppb) | Styrene (ppb) |
|---|---|---|---|---|
| 10% | 0.07 | 0.04 | 0.05 | 0.06 |
| 25% | 0.09 | 0.05 | 0.06 | 0.09 |
| 50% | 0.14 | 0.06 | 0.08 | 0.14 |
| 75% | 0.20 | 0.09 | 0.12 | 0.23 |
| 90% | 0.28 | 0.14 | 0.16 | 0.34 |
| average | 0.16 | 0.09 | 0.10 | 0.18 |

Table S2 Average nitrate concentrations at different sites during winter and summer and the corresponding references.

| Winter site | Measurement periods | NO$_3^-$ (μg/m$^3$) | Provenance | Summer site | Measurement periods | NO$_3^-$ (μg/m$^3$) | Provenance |
|---|---|---|---|---|---|---|---|
| Handan | 2015 | 26.18 | (Li et al., 2017) | Yucheng | 2013/2014 | 18 | (Wen et al., 2018) |
| Tianjing | 2013 | 29.4 | (Zou et al., 2018) | Beijing | 2011 | 16.8 | (Yang et al., 2017) |
| Beijing | 2013 | 16.4 | (Hu et al., 2017) | Jinan | 2008 | 19.2 | (Gao et al., 2011) |
| Tianjing | 2003 | 25.5 | (Cao et al., 2012) | Beijing | 2005 | 13.7 | (Pathak and Wu, 2009) |
| Beijing | 2003 | 13.1 | (Cao et al., 2012) | Beijing | 2003 | 9.9 | (Cao et al., 2012) |
| Nanjing | 2014/2015 | 22.3 | (This study) | Nanjing | 2014/2015 | 11.8 | (This study) |
| Shanghai | 2013 | 22.5 | (Zhang et al., 2015) | Shanghai | 2005 | 7.1 | (Pathak and Wu, 2009) |
| Linan | 2013 | 15 | (Zhang et al., 2015) | Hangzhou | 2003 | 5.5 | (Cao et al., 2012) |
| Shanghai | 2003 | 17.5 | (Cao et al., 2012) | Shanghai | 2003 | 2.6 | (Cao et al., 2012) |
| Hong Kong | 2011/2012 | 3.3 | (Griffith et al., 2015) | Hong Kong | 2011/2012 | 1.1 | (Griffith et al., 2015) |
| Hong Kong | 2011 | 3.84 | (Bian, et al., 2014) | Hong Kong | 2011 | 0.53 | (Bian, et al., 2014) |
| Zhongshan | 2010 | 8.496 | (Wang et al., 2012) | Shenzhen | 2009 | 4.45 | (He et al., 2011) |
| Guangzhou | 2003 | 11.5 | (Cao et al., 2012) | Guangzhou | 2003 | 1.2 | (Cao et al., 2012) |

Table S3 The seasonal and annual average mass concentrations of PM$_{2.5}$, ammonium and other related gases at SORPES station from 2014.3 to 2016.2.

|  | NH$_4^+$ ($\mu$g/m$^3$) | PM$_{2.5}$ ($\mu$g/m$^3$) | NOx ppb | O$_3$ ppb | NH$_3$ ($\mu$g/m$^3$) | NO$_3$/PM$_{2.5}$ |
|---|---|---|---|---|---|---|
| Jan. | 14.8 | 93.8 | 46.7 | 12.1 | 5.0 | 23.7% |
| Feb. | 12.9 | 78.6 | 28.3 | 22.8 | 6.2 | 26.3% |
| Mar. | 11.0 | 67.7 | 28.5 | 24.0 | 5.3 | 26.4% |
| Apr. | 9.3 | 58.9 | 23.0 | 31.6 | 5.8 | 24.5% |
| May | 8.9 | 67.2 | 22.9 | 39.1 | 9.2 | 19.4% |
| Jun. | 11.2 | 70.2 | 18.4 | 35.4 | 10.9 | 22.6% |
| Jul. | 9.6 | 48.6 | 19.9 | 29.3 | 14.9 | 20.3% |
| Aug. | 7.8 | 41.9 | 15.3 | 28.8 | 8.5 | 18.5% |
| Sept. | 6.6 | 41.1 | 20.0 | 28.5 | 6.0 | 18.6% |
| Oct. | 8.6 | 61.8 | 30.7 | 28.2 | 6.5 | 21.1% |
| Nov. | 10.7 | 70.9 | 33.6 | 15.0 | 4.4 | 25.7% |
| Dec. | 13.1 | 88.8 | 47.0 | 9.6 | 4.9 | 23.4% |
| Annual | 10.5 | 65.8 | 27.9 | 24.8 | 7.4 | 22.6% |

**References**

[revised manuscript text omitted]